EMBO
Molecular Medicine

# Administration of RANKL boosts thymic regeneration upon bone marrow transplantation

Noella Lopes[1], Hortense Vachon[1], Julien Marie[2,3] & Magali Irla[1,*]

## Abstract

Cytoablative treatments lead to severe damages on thymic epithelial cells (TECs), which result in delayed *de novo* thymopoiesis and a prolonged period of T-cell immunodeficiency. Understanding the mechanisms that govern thymic regeneration is of paramount interest for the recovery of a functional immune system notably after bone marrow transplantation (BMT). Here, we show that RANK ligand (RANKL) is upregulated in CD4[+] thymocytes and lymphoid tissue inducer (LTi) cells during the early phase of thymic regeneration. Importantly, whereas RANKL neutralization alters TEC recovery after irradiation, *ex vivo* RANKL administration during BMT boosts the regeneration of TEC subsets including thymic epithelial progenitor-enriched cells, thymus homing of lymphoid progenitors, and *de novo* thymopoiesis. RANKL increases specifically in LTi cells, lymphotoxin α, which is critical for thymic regeneration. RANKL treatment, dependent on lymphotoxin α, is beneficial upon BMT in young and aged individuals. This study thus indicates that RANKL may be clinically useful to improve T-cell function recovery after BMT by controlling multiple facets of thymic regeneration.

**Keywords** bone marrow transplantation; lymphotoxin α; receptor activator of nuclear factor kappa B ligand; T-cell reconstitution; thymic regeneration
**Subject Category** Immunology

## Introduction

The thymus controls the generation of a diverse repertoire of T lymphocytes. Cortical thymic epithelial cells (cTECs) support the differentiation of T-cell progenitors and the conversion of CD4[+]CD8[+] double-positive (DP) thymocytes into CD4[+]CD8[−] and CD4[−]CD8[+] single-positive (SP) cells (Anderson & Takahama, 2012). Medullary TECs (mTECs) purge the TCR repertoire of hazardous autoreactive T cells by expressing thousands of tissue-restricted antigens (TRAs), which are controlled by Aire (*Autoimmune Regulator*) and Fezf2 (*Forebrain Expressed Zinc Finger 2*) factors (Derbinski

*et al*, 2001; Anderson *et al*, 2002; Sansom *et al*, 2014; Takaba *et al*, 2015). Reciprocally, thymocytes sustain TEC differentiation and organization. This complex interplay is referred to as thymic cross talk (van Ewijk *et al*, 1994; Lopes *et al*, 2015).

Cytoablative treatments such as radiation or chemotherapy, used to prepare patients notably to bone marrow (BM) transplantation (BMT), severely affect not only hematopoietic cells but also TECs, which results in delayed T-cell reconstitution (Adkins *et al*, 1988; van den Brink *et al*, 2004; Fletcher *et al*, 2009; Hollander *et al*, 2010). Thymic injury triggered by total body irradiation (TBI) leads to profound alterations characterized by a drastic reduction in the cortex resulting from the massive depletion of DP thymocytes and a significant decrease in the medulla (Irla *et al*, 2013). Alterations in TEC ultrastructure and a reduction in some stromal cells have also been reported (Adkins *et al*, 1988; Irifune *et al*, 2004).

The recovery of a fully competent T-cell compartment is therefore a prolonged process that is considerably delayed compared to that of myeloid, NK or B cells (Hakim *et al*, 1997). This period of compromised immunity is prompt to serious clinical consequences such as opportunistic infections, autoimmunity, or tumor relapse and could lead to post-transplant morbidity and mortality (Curtis *et al*, 1997; Small *et al*, 1999; van den Brink *et al*, 2004; King *et al*, 2004; Parkman *et al*, 2006). Identifying an effective treatment that acts at several levels by improving (i) the regeneration of thymic epithelial progenitor cells (TEPCs), which are critical for the renewal of stromal niches; (ii) the recovery of cTECs and mTECs, which control the different steps of thymopoiesis; and (iii) thymus homing of T-cell progenitors, which is important for T-cell recovery, is of paramount clinical interest to optimally boost thymic regeneration (Penit & Ezine, 1989; Chen *et al*, 2004; Zlotoff *et al*, 2011).

RANK ligand, a TNF family member, has emerged as an important regulator of epithelial cell growth and differentiation in different tissues such as mammary glands during pregnancy (Fata *et al*, 2000), the renewal and epidermal growth of the hair follicles (Duheron *et al*, 2011), or M-cell differentiation from intestinal epithelial cells in Payer's patches (Knoop *et al*, 2009). In the embryonic thymus, RANKL provided by a subset of innate lymphoid cells, namely lymphoid tissue inducers (LTi) cells, and invariant Vγ5[+]TCR[+] T-cell progenitors promotes the emergence of Aire[+] mTECs, which mediate T-cell tolerance (Anderson *et al*, 2002; Rossi *et al*, 2007; Roberts *et al*, 2012). In the postnatal thymus, RANKL

1  Centre d'Immunologie de Marseille-Luminy, Aix Marseille Université, INSERM, CNRS, Marseille Cedex 09, France
2  Department of Immunology Virology and Inflammation, Cancer Research Center of Lyon (CRCL) UMR INSERM1052, CNRS 5286, Lyon, France
3  TGF-b and Immune Evasion, Tumor Immunology Program, DKFZ, Heidelberg, Germany
  *Corresponding author. Tel: +33 4 91 26 94 97; Fax: +33 4 91 26 94 30; E-mail: magali.irla@inserm.fr

  

produced by SP thymocytes enhances numbers of Aire$^+$ mTECs and the size of the medulla (Akiyama et al, 2008; Hikosaka et al, 2008; Irla et al, 2008, 2010; Ohigashi et al, 2011; Lopes et al, 2015). However, although RANKL is a potent inducer of mTEC differentiation at steady state, whether and how RANKL drives thymic regeneration upon BMT remain unknown.

Here, we show that RANKL is upregulated early after thymic damage in CD4$^+$ thymocytes and LTi cells. Importantly, whereas the in vivo neutralization of RANKL alters TEC regeneration after TBI, the ex vivo administration of RANKL substantially enhances the cellularity of cTEC and mTEC subsets as well as TEPC-enriched cells. Furthermore, we show that RANKL treatment induces lymphotoxin α (LTα) upregulation specifically in LTi cells, which express its cognate receptor, RANK. Although at steady state LTα$^{-/-}$ mice show normal TEC subsets, Aire$^+$ mTEC differentiation and T-cell development (De Togni et al, 1994; Venanzi et al, 2007; Seach et al, 2008), we demonstrate that LTα is critical for TEC regeneration and T-cell reconstitution. Importantly, RANKL administration during the early phase of BMT boosts the regeneration of TEPCs and TEC subsets, ameliorates T-cell progenitor homing and de novo thymopoiesis, which enhances peripheral T-cell reconstitution. Furthermore, we show that the effects mediated by RANKL depend on LTα expression and are also beneficial upon BMT in mice with early thymic involution. Altogether, our findings identify that the administration of RANKL constitutes a new therapeutic strategy to boost thymic regeneration upon BMT by acting at several levels: TEC recovery, T-cell progenitor homing, and de novo thymopoiesis.

# Results

## RANKL is upregulated during the early phase of thymic regeneration

Because at steady state RANKL has been reported as a potent regulator of mTEC differentiation (Rossi et al, 2007; Hikosaka et al, 2008; Ohigashi et al, 2011), we investigated whether this cytokine plays a role in thymic regeneration. To this, we first analyzed RANKL expression in the thymus at day 3 after SL-TBI (d3 SL-TBI) and found that RANKL was substantially upregulated in CD45$^+$ hematopoietic cells compared to untreated (UT) WT mice (Fig 1A). We next investigated the cell identity of hematopoietic cells that overexpressed RANKL. We found that recipient CD4$^+$ thymocytes and LTi cells, both previously described to be partially radio-resistant (Ueno et al, 2004; Dudakov et al, 2012; Fig 1B and C, Table 1, and Appendix Fig S1), upregulated RANKL at d3 SL-TBI (Fig 1D). Strikingly, although LTi cells represent a small subset in the thymus, they expressed higher levels of RANKL than CD4$^+$ thymocytes after TBI and upregulated RANKL in a radiation dose-dependent manner (Fig 1D). To analyze the contribution of CD4$^+$ thymocytes, we analyzed Rankl expression in the thymus of ZAP-70$^{-/-}$ mice, lacking SP thymocytes (Negishi et al, 1995; Kadlecek et al, 1998). At d3 SL-TBI, while the expression of Rankl mRNA was strongly upregulated in the WT thymus, no detectable increase of Rankl mRNA was observed in irradiated ZAP-70$^{-/-}$ thymus (Fig 1E). These results indicate that CD4$^+$ thymocytes are crucial for RANKL upregulation after TBI, which in line with their high numbers after irradiation (Table 1). Since ZAP-70$^{-/-}$ mice have normal DP cells, these results

also indicate that DP cells are not involved in RANKL upregulation. Given that LTi cells expressed high levels of RANKL after irradiation (Fig 1D), we decided to further define the contribution of this cell type in RANKL expression by analyzing the thymus from Rorc$^{-/-}$ mice, defective in LTi cells (Sun et al, 2000; Eberl et al, 2004). Irradiated Rorc$^{-/-}$ mice failed to increase Rankl, suggesting that this cell type was implicated in RANKL upregulation (Fig 1E). However, Rorc$^{-/-}$ mice were described to show reduced numbers of CD4$^+$ thymocytes (Sun et al, 2000). In order to clearly determine the contribution of LTi cells, we then analyzed RANKL expression in the thymus of Rag2$^{-/-}$ mice, showing an early block in T-cell development at the double-negative DN3 stage but exhibiting LTi cells (Shinkai et al, 1992). At d3 SL-TBI, Rankl mRNA was upregulated in the Rag2$^{-/-}$ thymus but at lesser extent than in WT thymus, confirming that LTi cells also contribute to RANKL overexpression after TBI (Fig 1E). Interestingly, RANKL was upregulated in CD4$^+$ SP and LTi cells until day 10 after SL-TBI with no hematopoietic rescue (Fig 1F). Of note, LTi cell ability to produce high level of RANKL in response to SL-TBI was much more pronounced than that of CD4$^+$ thymocytes. Altogether, these data indicate that RANKL is naturally upregulated in both CD4$^+$ SP and LTi cells at the early phase of thymic regeneration.

## RANKL neutralization inhibits TEC regeneration whereas ex vivo RANKL administration boosts TEC recovery after irradiation

The aforementioned data strongly suggest that RANKL could play a role in thymic regeneration after irradiation. To confirm this assumption, WT mice were treated with a neutralizing anti-RANKL antibody (IK22/5) during 3 days after SL-TBI. PBS- and isotype antibody-treated mice were used as controls. RANKL neutralization was sufficient to prevent TEC regeneration illustrated by a 2.5-fold decrease in numbers of total TECs (EpCAM$^+$), cTECs (EpCAM$^+$UEA-1$^-$Ly51$^+$), and mTECs (EpCAM$^+$UEA-1$^+$Ly51$^-$) compared to controls (Fig 2A). In addition, RANKL neutralization resulted in a decrease in CD80$^{hi}$Aire$^-$ and CD80$^{hi}$Aire$^+$ mTECs as well as of several TEC subsets identified by MHCII expression level (Wong et al, 2014), including cTEC$^{hi}$ (MHCII$^{hi}$UEA-1$^-$), mTEC$^{hi}$ (MHCII$^{hi}$ UEA-1$^+$), and mTEC$^{lo}$ (MHCII$^{lo}$UEA-1$^+$) (Fig 2A and B). Interestingly, a TEC population described to be enriched in TEPCs defined as α6-integrin$^{hi}$Sca-1$^{hi}$ in the TEC$^{lo}$ (MHCII$^{lo}$UEA-1$^{lo}$) subset (Wong et al, 2014) was also decreased (Fig 2C). In a therapeutic perspective, we next investigated whether conversely the ex vivo administration of RANKL protein could improve TEC regeneration. WT mice were treated with RANKL-GST protein during 3 days after SL-TBI. PBS- and GST-treated mice were used as controls. Remarkably, RANKL-treated mice showed a 2-fold increase in numbers of total TECs, cTECs, and mTECs compared to controls (Fig 2A). RANKL treatment also enhanced CD80$^{hi}$Aire$^-$ and CD80$^{hi}$Aire$^+$ mTECs as well as cTEC$^{hi}$, mTEC$^{hi}$, TEC$^{lo}$, mTEC$^{lo}$, and TEPC-enriched cells (Fig 2B and C).

To gain mechanistic insights into the mode of action of RANKL, we then analyzed the proliferation of cTECs, mTECs, and TEPC-enriched cells. Numbers of proliferating Ki-67$^+$ cells in these three subsets were decreased after RANKL neutralization, whereas they were increased after ex vivo RANKL administration (Fig 2D). Interestingly, the analysis of purified mTECs from RANKL-treated mice showed reduced expression of Bax, Bid, and Bak pro-apoptotic

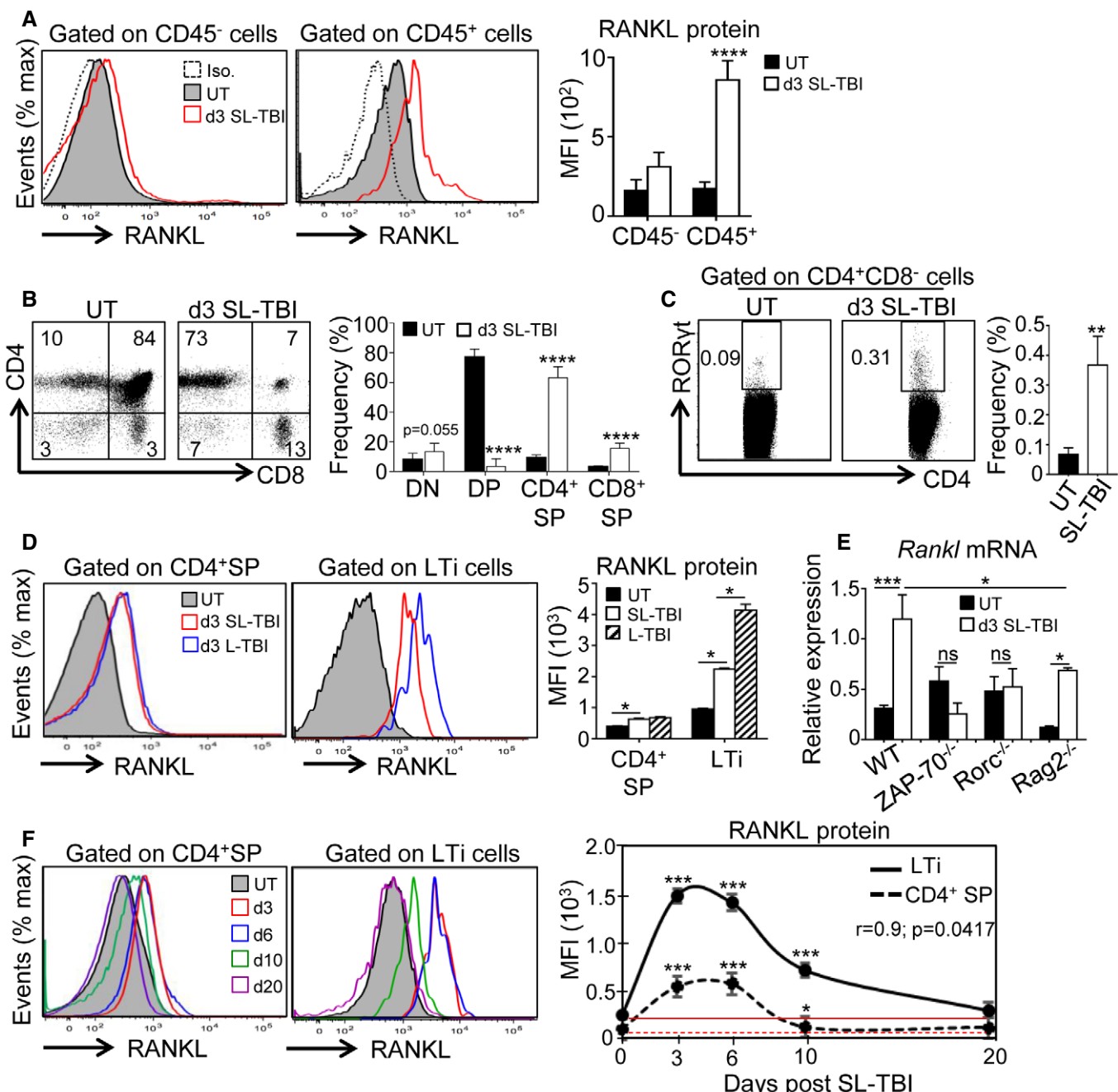

**Figure 1.  RANKL is upregulated in CD4+ SP and LTi cells during the course of thymic regeneration.**

A       Expression of RANKL protein analyzed by flow cytometry in CD45− and CD45+ thymic cells from untreated (UT) WT mice or at d3 SL-TBI.
B, C    Flow cytometry profiles and frequencies of DN (double negative), DP (double positive), CD4+ and CD8+ SP (single positive) (B), and LTi cells (C) from untreated (UT) WT mice or at d3 SL-TBI.
D       Expression level of RANKL protein in CD4+ SP and LTi cells from UT WT mice or at d3 SL-TBI and L-TBI.
E       Expression of *Rankl* mRNA in the total thymus isolated from UT WT, Rorc−/−, ZAP-70−/−, and Rag2−/− mice or at d3 SL-TBI (n = 3–6 mice per genotype).
F       CD4+ SP and LTi cells from UT WT mice or at d3, d6, d10, and d20 SL-TBI with no hematopoietic rescue were analyzed for the expression of RANKL protein. Mean fluorescence intensity (MFI) of RANKL in CD4+ SP and LTi cells over time following SL-TBI. The red lines represent the MFI of RANKL at baseline.

Data information: Data are shown as mean ± SEM and are pooled of four independent experiments with similar results (n = 3–4 mice per group). *P < 0.05; **P < 0.01; ***P < 0.001, ****P < 0.0001. Exact *P*-values and statistical tests used to calculate them are provided in Appendix Table S2.

genes and that of cTECs exhibited reduced expression of *Bax* as well as an increased expression of the *Bcl-xl* anti-apoptotic gene (Fig 2E). Furthermore, the density of medullary Aire+ cells and the

expression of *Aire* and *Aire*-dependent TRAs were also enhanced in response to RANKL (Fig 2F and G). Of note, we also found that RANKL stimulated in cTECs the expression of *Selp, Icam-1,* and

**Table 1. Cell numbers of lymphoid cells observed in the thymus of WT mice before and after d3 SL-TBI.**

| Cell types | Untreated | d3 SL-TBI | *P*-values | Category |
|---|---|---|---|---|
| DN ($\times 10^6$) | 15.86 ± 2.62 | 0.49 ± 0.07 | < 0.0001 | **** |
| DP ($\times 10^6$) | 130.70 ± 14.46 | 0.27 ± 0.07 | < 0.0001 | **** |
| CD4$^+$ SP ($\times 10^6$) | 18.01 ± 2.07 | 2.61 ± 0.20 | < 0.0001 | **** |
| CD8$^+$ SP ($\times 10^6$) | 5.02 ± 0.79 | 0.68 ± 0.05 | < 0.0001 | **** |
| Foxp3$^+$ Tregs ($\times 10^5$) | 3.5 ± 0.33 | 0.14 ± 0.01 | < 0.0001 | **** |
| LTi cells ($\times 10^3$) | 3.35 ± 0.72 | 1.58 ± 0.12 | 0.0021 | ** |

Average ± SD (Untreated: $n = 10$ thymi and d3SL-TBI: $n = 18$ thymi). *P*-values were calculated using Student's *t*-test. **$P < 0.01$; ****$P < 0.0001$.

*Ccl21*, implicated in the thymus homing of lymphoid progenitors (Fig 2H). Altogether, these data point out that RANKL controls endogenous TEC recovery and that *ex vivo* RANKL administration boosts TEC regeneration by enhancing their proliferation, survival, and differentiation after thymic damage.

### RANKL controls LTα expression specifically in LTi cells after TBI

We next investigated deeper the underlying mechanism(s) of RANKL treatment in the thymus after TBI. During embryogenesis, *in vitro* experiments have shown that RANKL induces LTα expression in peripheral LTi cells (Yoshida *et al*, 2002). A possible mechanism is thus that RANKL acts on this cell type. Interestingly, we found that irradiation led to the upregulation of RANKL cognate receptor, RANK on LTi cells (Fig 3A). Furthermore, to assess whether RANKL regulates LTα production in thymic LTi cells after irradiation, hematopoietic cells from irradiated WT thymus were stimulated *in vitro* with either RANKL-GST or GST. RANKL stimulation significantly upregulated LTα expression in LTi cells, while the addition of RANKL antagonist, RANK-Fc, fully abolished LTα induction, demonstrating the specificity of the treatment used (Fig EV1A). It is notable that among radio-resistant hematopoietic cells, only LTi cells upregulated LTα protein (Fig EV1B). In line with this, *in vivo* administration of RANKL during 3 days after SL-TBI also induced LTα upregulation selectively in thymic LTi cells (Fig 3B and C). Conversely, the administration of a neutralizing anti-RANKL antibody inhibited LTα upregulation only in LTi cells (Fig 3D and E). These data thus indicate that RANKL controls LTα upregulation specifically in LTi cells after SL-TBI. Moreover, LTi-deficient Rorc$^{-/-}$ thymus failed to increase *LT*α mRNA at d3 SL-TBI, suggesting that LTi cells are required for LTα overexpression after irradiation (Fig 3F). In contrast, LTα upregulation was maintained at WT level in the irradiated ZAP-70$^{-/-}$ thymus, lacking SP cells, indicating that radio-resistant CD4$^+$ thymocytes are not involved in LTα upregulation after SL-TBI (Fig 3F). Since DP thymocytes constitute a non-negligible number after irradiation even if they are massively eliminated (Table 1) and that ZAP-70$^{-/-}$ mice have normal DP cells, we analyzed their potential contribution in LTα upregulation using Rag2$^{-/-}$ mice (lacking DP and SP cells). Irradiated Rag2$^{-/-}$ thymus expressed LTα at a similar level than that of irradiated WT mice, indicating that DP cells are not involved in LTα expression after SL-TBI and confirming that LTi cells are the main providers (Fig 3G). Furthermore, the thymus of irradiated WT mice deprived before of DP cells by dexamethasone treatment (Purton *et al*, 2004), also exhibited the same LTα expression level than

that observed in irradiated WT and Rag2$^{-/-}$ mice (Fig 3G), confirming that LTα upregulation after SL-TBI does not rely on DP thymocytes. These data thus demonstrate that LTi cells are critical for LTα upregulation after irradiation.

Interestingly, LTα overexpression in LTi cells tightly correlated with that of RANKL during the course of BMT (Fig 3H). We found that LTα protein was selectively upregulated in LTi cells from recipient and not from donor origin until day 6 after BMT (Fig 3I and J), showing the importance of the host LTi cells in LTα production. Moreover, RANKL was expressed at normal level at d3 SL-TBI in LTα$^{-/-}$ mice, suggesting that LTα did not regulate RANKL (Fig 3K). We further observed that both *Lt*α and *Lt*β mRNAs were increased in the total thymus at d3 SL-TBI compared to UT WT mice (Fig 3L) and that LTα protein was specifically induced in hematopoietic cells (Fig 3M). We thus hypothesized that LTα could be expressed as a membrane anchored LTα1β2 heterocomplex, which only binds to LTβR (Gommerman & Browning, 2003). We used a soluble LTβR-Fc fusion protein, which detects the two LTβR ligands, LTα1β2 and LIGHT. In contrast to *Lt*α and *Lt*β, *Light* mRNA was slightly expressed and not upregulated after thymic injury (Appendix Fig S2), indicating that LTβR-Fc staining detects only LTα1β2 in LTi cells, which was upregulated in a radiation dose-dependent manner (Fig 3N and O). These data thus show that RANKL treatment induces LTαβ upregulation specifically in LTi cells early after thymic injury.

### LTα is critical for TEC regeneration and *de novo* thymopoiesis during the course of BMT

We next addressed whether LTα upregulation in response to RANKL treatment after TBI is involved in thymic regeneration. In line with this assumption, total TECs, cTECs, and mTECs but also TEPC-enriched cells upregulated LTβ receptor (LTβR) at d3 SL-TBI (Fig 4A). While at steady state, LTα$^{-/-}$ mice, in which the expression of LTα1β2 is fully lost, did not show any defect in TEC subsets (Fig 4B–F; Venanzi *et al*, 2007), numbers of cTECs, mTECs, and mTEC subsets (CD80$^{lo}$Aire$^-$, CD80$^{hi}$Aire$^-$, and CD80$^{hi}$Aire$^+$) as well as the density of Aire$^+$ cells were dramatically reduced at d3 SL-TBI (Fig EV2A–C). Of note, no significant defect in CD45$^-$PDGFRα$^+$ fibroblasts or in thymic LTi cells was observed in LTα$^{-/-}$ mice at d3 SL-TBI (Fig EV2D and E). To definitively address the role of LTα during thymic recovery after BMT, lethally irradiated CD45.2 WT or LTα$^{-/-}$ recipients were reconstituted with CD45.1 congenic BM cells (WT CD45.1:WT or WT CD45.1:LTα$^{-/-}$ mice) and TEC numbers were analyzed at days 10, 21, and 65 after BMT (Fig 4B–F). We observed reduced numbers of total TECs, cTECs, and mTECs as well

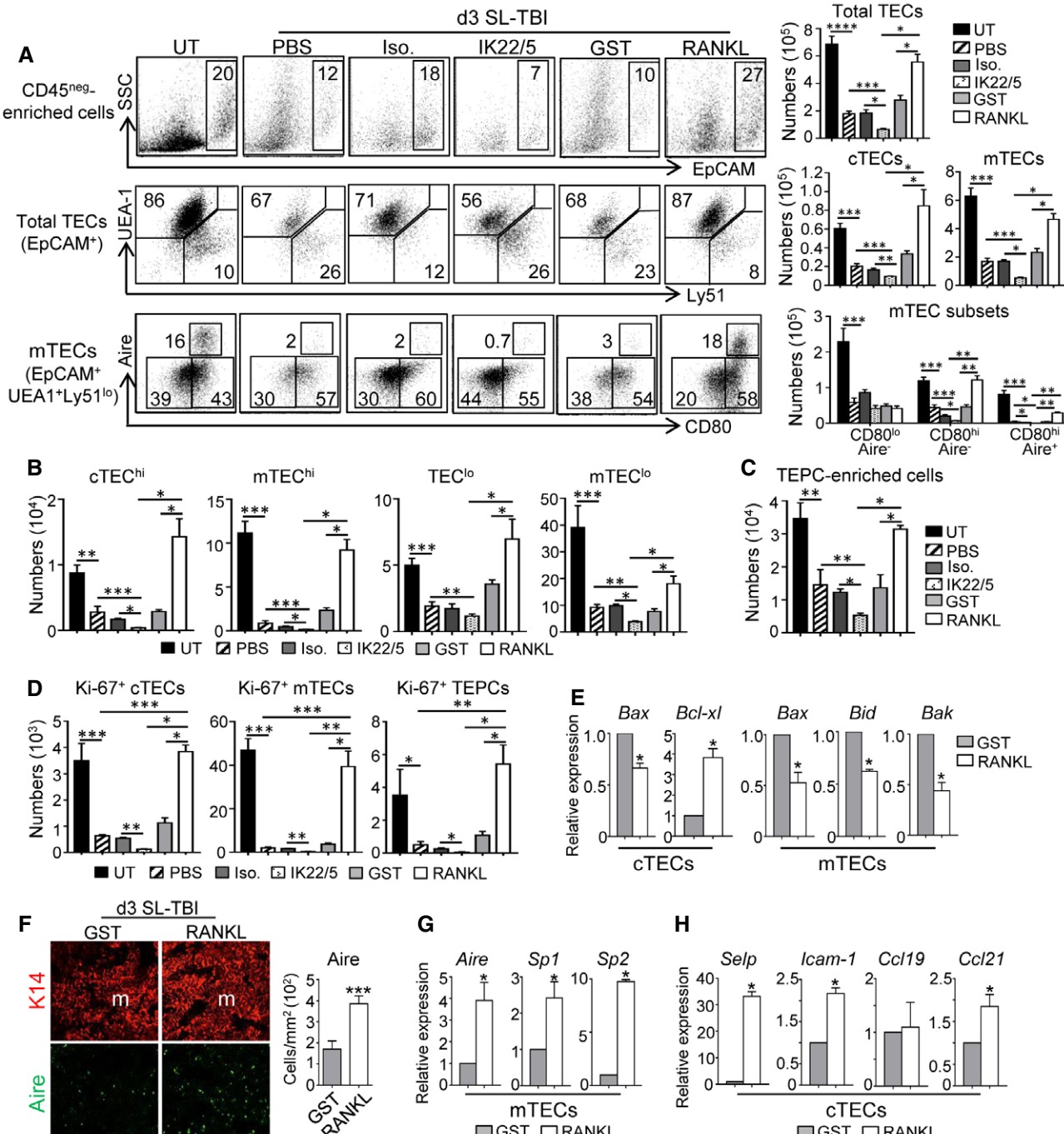

**Figure 2. RANKL is crucially involved in TEC regeneration after TBI.**

A   Flow cytometry profiles and numbers of total TECs (EpCAM$^+$), cTECs (UEA-1$^-$Ly51$^+$), mTECs (UEA-1$^+$Ly51$^-$), and mTEC subsets (CD80$^{lo}$Aire$^-$, CD80$^{hi}$Aire$^-$, and CD80$^{hi}$Aire$^+$) analyzed in CD45$^{neg}$-enriched cells by AutoMACS from UT WT mice or treated with PBS, a RANKL isotype control antibody (Iso.), a neutralizing anti-RANKL antibody (IK22/5), GST or RANKL proteins during 3 days upon SL-TBI.

B–D   Histograms show numbers of cTEC$^{hi}$ (MHCII$^{hi}$UEA-1$^-$), mTEC$^{hi}$ (MHCII$^{hi}$UEA-1$^+$), TEC$^{lo}$ (MHCII$^{lo}$UEA-1$^-$), mTEC$^{lo}$ (MHCII$^{lo}$UEA-1$^+$) (B); TEPC-enriched cells (defined as α6-integrin$^{hi}$Sca-1$^{hi}$ in the TEC$^{lo}$ subset) (C); and proliferating Ki-67$^+$ cTECs, mTECs, and TEPC-enriched cells (D).

E   Expression of mRNAs coding for pro- (*Bax*, *Bid*, *Bak*) and anti-apoptotic (*Bcl-xl*) proteins analyzed by qPCR in purified cTECs and mTECs.

F   Thymic sections from WT mice treated with GST or RANKL during 3 days upon SL-TBI were stained for the expression of K14 and Aire. The histogram shows the density of Aire$^+$ cells in medullary area. m denotes the medulla. Fifteen sections were quantified for each condition; scale bar: 100 μm.

G, H   The expression of *Aire* and TRAs (*Sp1* and *Sp2*) in purified mTECs (G) and *Selp, Icam-1, Ccl19,* and *Ccl21* in purified cTECs (H) from WT mice treated with GST or RANKL was analyzed by qPCR.

Data information: Data are shown as mean ± SEM and are pooled of three independent experiments with similar results (*n* = 3–4 mice per group). *P < 0.05; **P < 0.01; ***P < 0.001, ****P < 0.0001. Exact *P*-values and statistical tests used to calculate them are provided in Appendix Table S2.

**Figure 3.  RANKL administration induces LTα upregulation specifically in thymic LTi cells after TBI.**

A    Expression of RANK receptor in thymic LTi cells from UT WT ($n = 6$) mice and at d3 SL-TBI ($n = 6$).
B    Expression level of LTα protein in thymic LTi cells from WT mice treated *in vivo* with PBS ($n = 6$), GST ($n = 6$), or RANKL-GST ($n = 6$) during 3 days after SL-TBI.
C    LTα protein was analyzed in thymocyte subsets and LTi cells from WT mice treated *in vivo* with GST ($n = 9$) or RANKL-GST ($n = 9$) during 3 days after SL-TBI. Results are represented as fold change relative to the GST condition. Data are pooled of three experiments.
D    Expression level of LTα protein in thymic LTi cells from WT mice treated *in vivo* with PBS ($n = 6$), an isotype control (Iso.) ($n = 3$), or a neutralizing anti-RANKL antibody (IL22/5) ($n = 6$) during 3 days after SL-TBI.
E    LTα protein was analyzed in thymocyte subsets and LTi cells from WT mice treated *in vivo* with an isotype control ($n = 3$), or a neutralizing anti-RANKL antibody ($n = 6$) during 3 days after SL-TBI. Results are represented as fold change relative to the isotype condition.
F    Expression of *Ltα* mRNA in the total thymus isolated from UT WT, Rorc$^{-/-}$, and ZAP-70$^{-/-}$ mice or at d3 SL-TBI ($n = 3$–6 mice per genotype).
G    Expression of *Ltα* mRNA in the total thymus isolated from irradiated: WT, Rag2$^{-/-}$ mice, and WT mice treated 3 days before with dexamethasone (Dexa). Data are pooled of two to three experiments ($n = 6$–12 mice per group).
H    Correlation of RANKL and LTα expression in thymic LTi cells during the course of BMT. pBMT: post-bone marrow transplantation. Data are pooled of three independent experiments with similar results ($n = 3$–4 mice per group).
I    Expression level of LTα protein in thymic LTi cells from UT WT mice or at d3, d6, d10, and d21 after BMT. Data are pooled of three independent experiments with similar results ($n = 3$–4 mice per group).
J    Expression level of LTα protein analyzed by flow cytometry in thymic LTi cells from CD45.1 donor and CD45.2 host origin at d3 and d6 after BMT. Data are pooled of four experiments ($n = 3$–4 mice per group).
K    RANKL protein expression in LTi cells from WT and LTα$^{-/-}$ mice at d3 SL-TBI. Data are pooled of three experiments ($n = 3$–5 mice per group).
L    The expression of *Ltα* and *Ltβ* mRNAs was measured by qPCR in the total thymus from UT WT mice ($n = 4$) or at d3 SL-TBI ($n = 4$).
M    MFI of LTα protein in CD45$^-$ and CD45$^+$ thymic cells from UT WT ($n = 6$) mice or at d3 SL-TBI ($n = 6$).
N, O    Representative histogram of LTα (N) and LTβR-Fc staining (O) in LTi cells from UT WT mice ($n = 6$) or at d3 SL-TBI ($n = 6$) or L-TBI ($n = 6$).

Data information: Data are shown as mean ± SEM. *$P < 0.05$; **$P < 0.01$; ***$P < 0.001$, ****$P < 0.0001$. Exact *P*-values and statistical tests used to calculate them are provided in Appendix Table S2.

as mTEC subsets in WT CD45.1:LTα$^{-/-}$ mice compared to WT CD45.1:WT controls at all time points analyzed (Fig 4B–D). Moreover, cTEC$^{hi}$, mTEC$^{hi}$, TEC$^{lo}$, mTEC$^{lo}$, and TEPC-enriched cells were also reduced (Fig 4E and F). Importantly, total TECs, cTECs, mTECs, and TEPCs were less proliferative (Fig 4G). A reduced density of medullary Aire$^+$ cells was still detectable at d65 after BMT (Appendix Fig S3), and consequently, the expression of *Aire* and its dependent TRAs (*Sp1* and *Sp2*) were strongly affected (Fig 4H). The expression of an *Aire*-independent TRA (*casein β*) and *Fezf2* as well as its target genes (*Apoc3*, *Fabp9*, and *Resp18*) (Takaba *et al*, 2015) was also reduced. These data thus reveal that LTα is critical for TEC regeneration including TEPC-enriched cells during the course of BMT.

In line with these thymic environmental defects, thymocytes were reduced from DN to SP stage in WT CD45.1:LTα$^{-/-}$ mice after BMT (Fig 5A and B). Consequently, numbers of peripheral CD4$^+$ and CD8$^+$ T cells as well as CD4$^+$Foxp3$^+$ regulatory T cells (Tregs) from CD45.1 donor origin were reduced in the blood and spleen of WT CD45.1:LTα$^{-/-}$ mice from d21 to d100 after BMT (Fig EV3A–D). Furthermore, CD62L$^+$CD44$^-$ naïve CD4$^+$ and CD8$^+$ T cells were also decreased in the spleen of these mice (Fig EV3E). Consistently with these data, the detection of signal joint TCR excision circles (sjTRECs) was also reduced in purified peripheral CD4$^+$ and CD8$^+$ T cells (Fig EV3F), confirming that thymic activity was altered in BM-transplanted LTα$^{-/-}$ mice.

Since *de novo* thymopoiesis was impaired from the DN1 stage in WT CD45.1:LTα$^{-/-}$ mice (Fig 5B), we analyzed early T-lineage progenitors (ETPs; CD4$^-$CD8$^-$CD44$^+$CD25$^-$Lin$^-$CD117$^+$). Whereas numbers of ETPs were normal in LTα$^{-/-}$ thymus at steady state, ETPs from CD45.1 donor origin were reduced in WT CD45.1: LTα$^{-/-}$ chimeras until 2 months after BMT (Fig 5C). This defect was not attributable to impaired hematopoietic progenitors because normal numbers of prethymic progenitors were observed in the BM of these mice (Fig 5C). We hypothesized that reduced ETPs could be due to a reduced homing capacity of circulating T-cell

progenitors. Thymus homing is controlled by a multistep adhesion cascade initiated by P-selectin slowing down T-cell progenitors and allowing them to respond to CCL25, CCL21/19 gradients and to engage with ICAM-1 and VCAM-1 expressed by the thymic stroma, leading to a firm arrest (Rossi *et al*, 2005; Scimone *et al*, 2006; Krueger *et al*, 2010; Zlotoff *et al*, 2010). We found that purified TECs from WT CD45.1:LTα$^{-/-}$ mice showed a reduced expression of *Ccl19* and *Ccl21* mRNAs at d21 after BMT, whereas purified fibroblasts from these mice displayed a decrease expression of *Ccl21* mRNA (Fig 5D). Furthermore, purified endothelial cells exhibited a reduced expression of *Icam-1*, *Vcam-1*, and *Selp* adhesion molecules (Fig 5D). TECs, fibroblasts and endothelial cells were thus defective in key molecules involved in thymus homing in BM-transplanted LTα$^{-/-}$ mice. To firmly demonstrate that thymus homing of T-cell progenitors was altered in LTα$^{-/-}$ mice, short-term homing assays were performed by injecting CD45.1 congenic BM cells into irradiated WT and LTα$^{-/-}$ recipients. LTα$^{-/-}$ thymus imported 3-fold less ETPs than WT thymus after thymic injury (Fig 5E). It is noteworthy that the effects of LTα on TEC and T-cell recovery were independent of those mediated by IL-23-regulated IL-22 described to be involved in thymic regeneration (Dudakov *et al*, 2012) since LTα$^{-/-}$ mice exhibited similar expression of these two cytokines after TBI (Fig 5F and G). Altogether, these results reveal that RANKL-regulated LTα constitutes an indispensable pathway for thymic regeneration.

## RANKL administration enhances thymic regeneration upon BMT in an LTα-dependent manner

Since RANKL treatment improves TEC regeneration upon SL-TBI (Fig 2), we next evaluated whether RANKL boosts thymic recovery during the course of BMT. WT mice transplanted with CD45.1 BM cells were treated with RANKL-GST or GST at d2, d4, and d6 after BMT, and thymic regeneration was analyzed at d21 (Fig 6A) and d65 (Appendix Fig S4A) after BMT. In these experiments, as observed at d3 SL-TBI (Fig 3B), RANKL treatment also upregulated

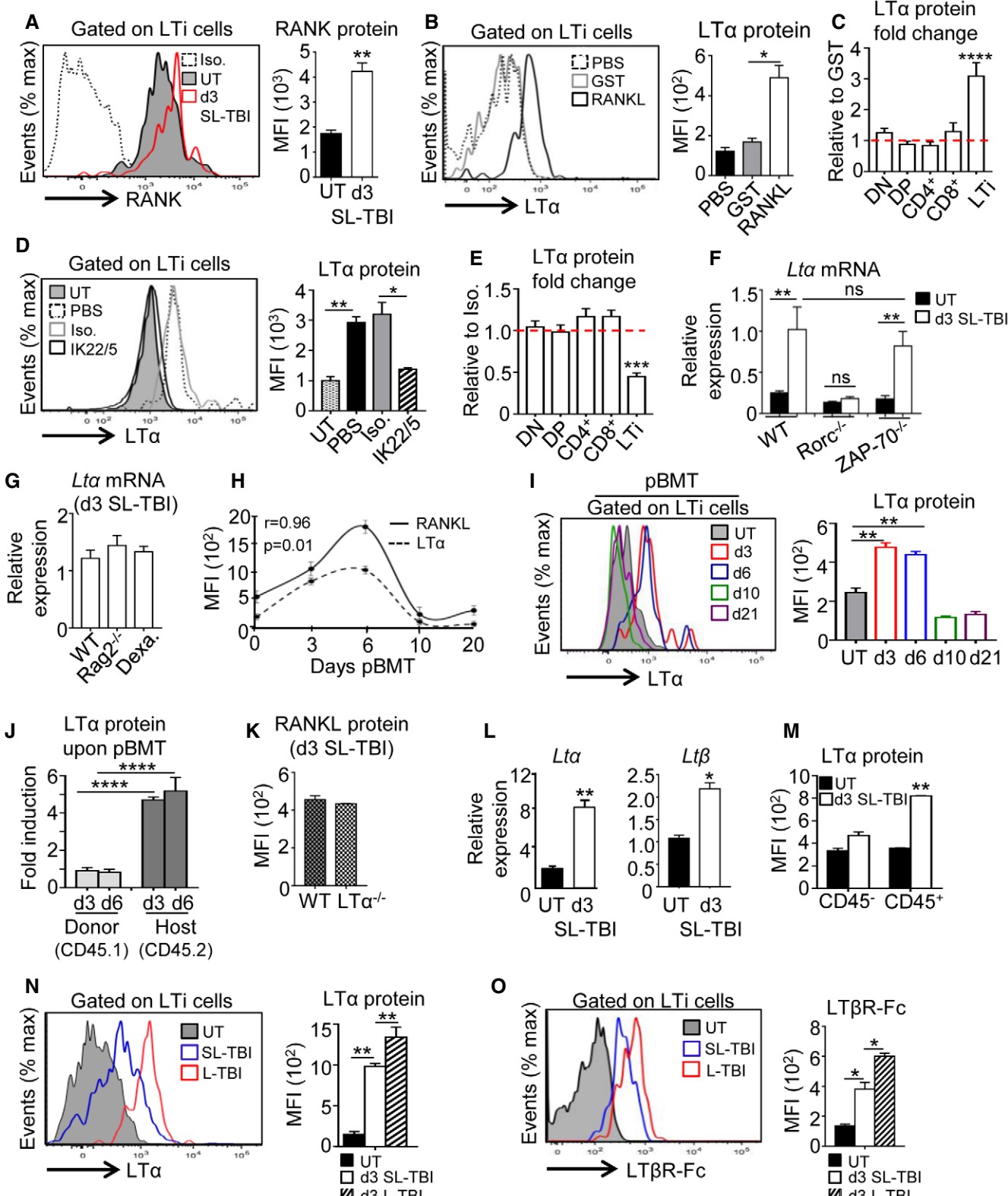

Figure 3.

LTα in LTi cells, which was still detectable at d21 pBMT (Fig 6B). RANKL-treated WT CD45.1:WT mice showed increased medullary areas, numbers of TEC subsets, Aire+ mTEC frequency, and

Aire-dependent TRAs compared to GST-treated mice (Fig 6C–F; Appendix Fig S4B). RANKL also increased numbers of CD31+ endothelial cells (Appendix Fig S5). Because we found that RANKL

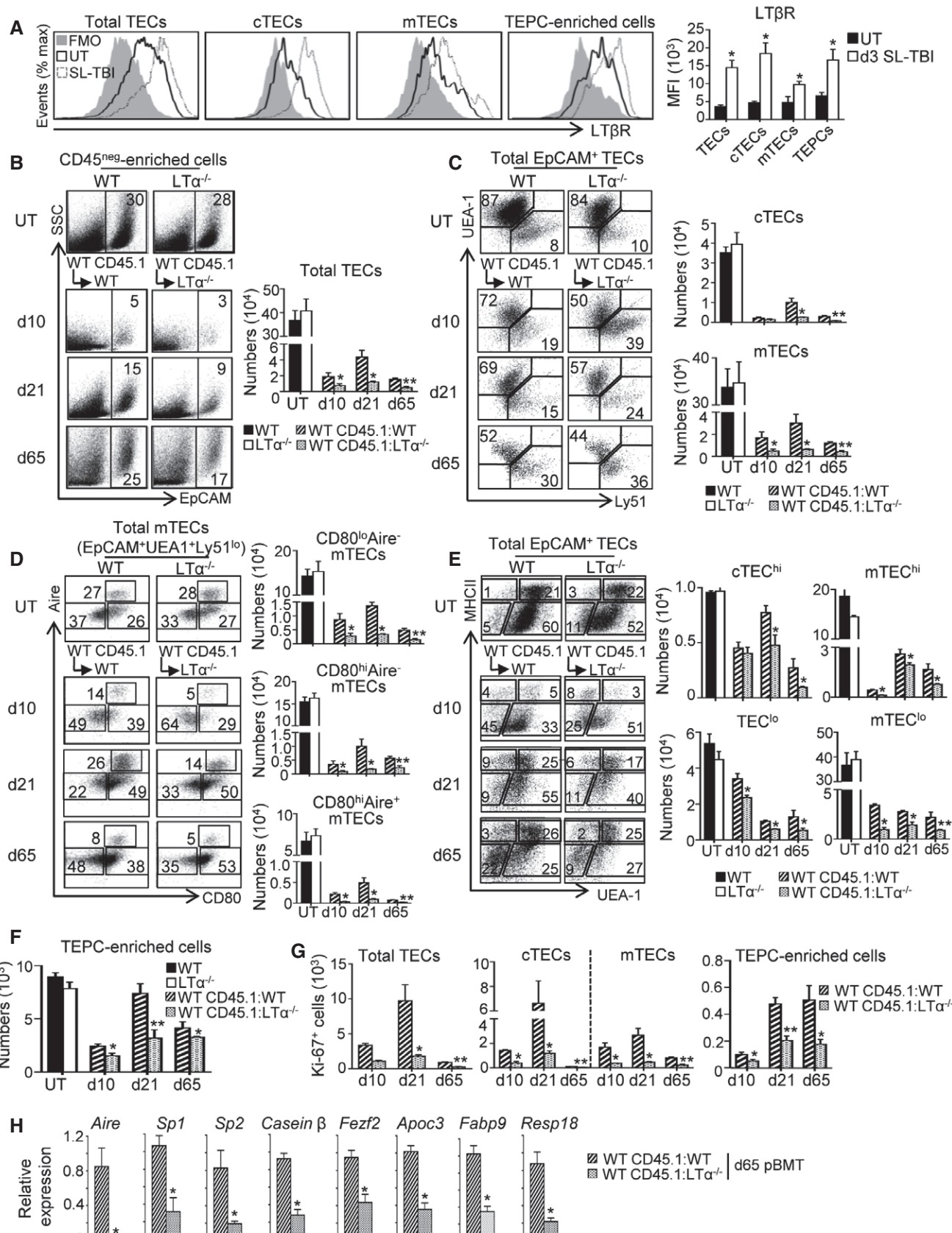

**Figure 4.**

◀

**Figure 4. LTα is critical for TEC regeneration during the course of BMT.**

A  Expression level of LTβR protein in total TECs, cTECs, mTECs, and TEPC-enriched cells from the thymus of UT WT mice (*n* = 6) and at d3 SL-TBI (*n* = 6) was analyzed by flow cytometry. FMO: Fluorescence Minus One.

B–F  Flow cytometry profiles and numbers of total TECs (B); cTECs, mTECs (C); mTEC subsets (D); cTEC^hi, mTEC^hi, TEC^lo, mTEC^lo (E); and TEPC-enriched cells (F) were analyzed in CD45^neg-enriched cells by AutoMACS from the thymus of UT WT and LTα^−/− mice or in WT CD45.1:WT and WT CD45.1:LTα^−/− chimeras at d10, d21, and d65 upon BMT.

G  Numbers of total proliferating Ki-67^+ TECs, cTECs, mTECs, and TEPC-enriched cells at the indicated time points.

H  The expression of mRNAs coding for *Aire*, *Aire*-induced TRAs (*Sp1* and *Sp2*); *Aire*-independent TRA (*casein β*); *Fezf2* and *Fezf2*-induced TRAs (*Apoc3*, *Fabp9*, and *Resp18*) was measured by qPCR in CD45^− thymic stromal cells from WT CD45.1:WT and WT CD45.1:LTα^−/− mice at d65 after BMT. Significance relative to WT CD45.1:WT chimeras.

Data information: Data are shown as mean ± SEM and are pooled of three independent experiments with similar results (*n* = 3–5 mice per group). *P < 0.05; **P < 0.01. Exact P-values and statistical tests used to calculate them are provided in Appendix Table S2.

regulates LTα (Fig 3B), we next investigated whether these beneficial effects on TECs mediated by RANKL require LTα expression. The administration of RANKL in WT CD45.1:LTα^−/− mice increased TEC numbers but to a lesser extent compared to RANKL-treated WT CD45.1:WT mice (Fig 6D; Appendix Fig S4B). In contrast, RANKL treatment in these mice did not enhance the size of the medulla, neither Aire^+ mTEC frequency nor *Aire*-dependent TRAs, indicating that LTα is critical for the regeneration of Aire^+ mTECs mediated by RANKL (Fig 6C, E and F).

Interestingly, RANKL treatment in WT CD45.1:WT chimeras substantially increased numbers of total donor cells and thymocytes of CD45.1 origin from ETP to SP stages at d21 and d65 upon BMT (Fig 6G; Appendix Fig S4C). In contrast, RANKL administration in WT CD45.1:LTα^−/− mice had a poor effect on *de novo* thymopoiesis. To decipher the mode of action of RANKL on T-cell reconstitution, we performed short-term homing assays in irradiated WT and LTα^−/− mice treated with GST or RANKL. Strikingly, the receptivity capacity of circulating progenitors was substantially enhanced in RANKL-treated WT CD45.1:WT mice compared to GST-treated controls (Fig 6H). Importantly, this was not due to increased numbers of prethymic progenitors in the BM upon RANKL treatment (Fig 6H). In contrast, RANKL had no effect on ETP homing in WT CD45.1:LTα^−/− mice. Consequently, RANKL treatment increased peripheral T-cell reconstitution only in WT CD45.1:WT mice after BMT (Fig EV4). Altogether, these data demonstrate that LTα is critical for optimal effects of RANKL administration on TEC regeneration, thymus homing of lymphoid progenitors, and T-cell reconstitution upon BMT.

Since the effects of RANKL-regulated LTα are independent of those mediated by IL-22 described to be involved in thymic regeneration (Fig 5F and G; Dudakov *et al*, 2012), we next investigated the respective efficiency of IL-22 and RANKL in thymic regeneration during BMT. To this, WT mice transplanted with CD45.1 BM cells were treated with IL-22, RANKL or both at d2, d4, and d6 after BMT and thymic regeneration was analyzed at d21. Whereas the concomitant administration of IL-22 and RANKL did not ameliorate thymic recovery compared to UT mice (Appendix Fig S6), we found that IL-22 and RANKL administrated alone increased similarly numbers of developing T cells including ETPs and ameliorated peripheral T-cell reconstitution (Appendix Fig S6A–C). Interestingly, IL-22 alone increased preferentially numbers of mTEC^hi including CD80^hiAire^− and CD80^hi Aire^+ mTEC subsets (Appendix Fig S6E and F), whereas RANKL alone enhanced numbers of all TEC populations including cTECs and mTECs (Appendix Fig S6D and E). The numbers of cTEC^hi, mTEC^hi,

TEC^lo, mTEC^lo, and TEPC-enriched cells were also increased in RANKL-treated mice (Appendix Fig S6F and G). In addition to IL-22, RANKL thus constitutes a new therapy to enhance thymic regeneration upon BMT by ameliorating both TEC and T-cell reconstitution.

### RANKL treatment also ameliorates thymic recovery upon BMT in aged individuals

Because the recovery of T-cell functions upon BMT is known to be delayed and less efficient in elderly patients compared to young individuals (Toubert *et al*, 2012), we finally investigated whether RANKL beneficial effects are persistent with age. To this, WT mice of 6–8 months, in which early thymic involution is characterized by a decline in TEC cellularity (Gray *et al*, 2006; Ki *et al*, 2014), were subjected to the same treatment described in Fig 6A. We found that RANKL increased numbers of cTECs, mTECs, and TEPC-enriched cells (Fig 7A and B). All thymocytes including ETPs were also increased in these mice (Fig 7C and D). Importantly, RANKL administration in WT CD45.1:LTα^−/− chimeras did not improve significantly TEC cellularity compared to RANKL-treated WT CD45.1:WT mice (Fig 7A and B). Moreover, RANKL treatment had only minor effects on *de novo* thymopoiesis in WT CD45.1:LTα^−/− chimeras (Fig 7C and D). Peripheral T-cell reconstitution was thus only enhanced in RANKL-treated WT CD45.1:WT mice (Appendix Fig S7). This set of data is consistent with the fact that LTi cells persisted with age and also upregulated LTα1β2 after TBI (Fig EV5A and B). Furthermore, BM-transplanted LTα^−/− mice of 6–8 months of age showed defective TEC regeneration, *de novo* thymopoiesis, and peripheral T-cell reconstitution (Fig EV5C–K). Altogether, our data indicate that RANKL treatment boosts thymic recovery after BMT not only in young but also in older individuals in an LTα-dependent manner.

## Discussion

Pre-BMT conditioning induces severe damages on the thymic microenvironment, which results in delayed lymphocyte production. It is therefore of paramount clinical interest to discover new molecules that enhance thymic regeneration for an efficient recovery of the immune system (van den Brink *et al*, 2004; Hollander *et al*, 2010).

Our study demonstrates that RANKL plays a crucial role in thymic recovery during BMT. Although that LTi cells strongly

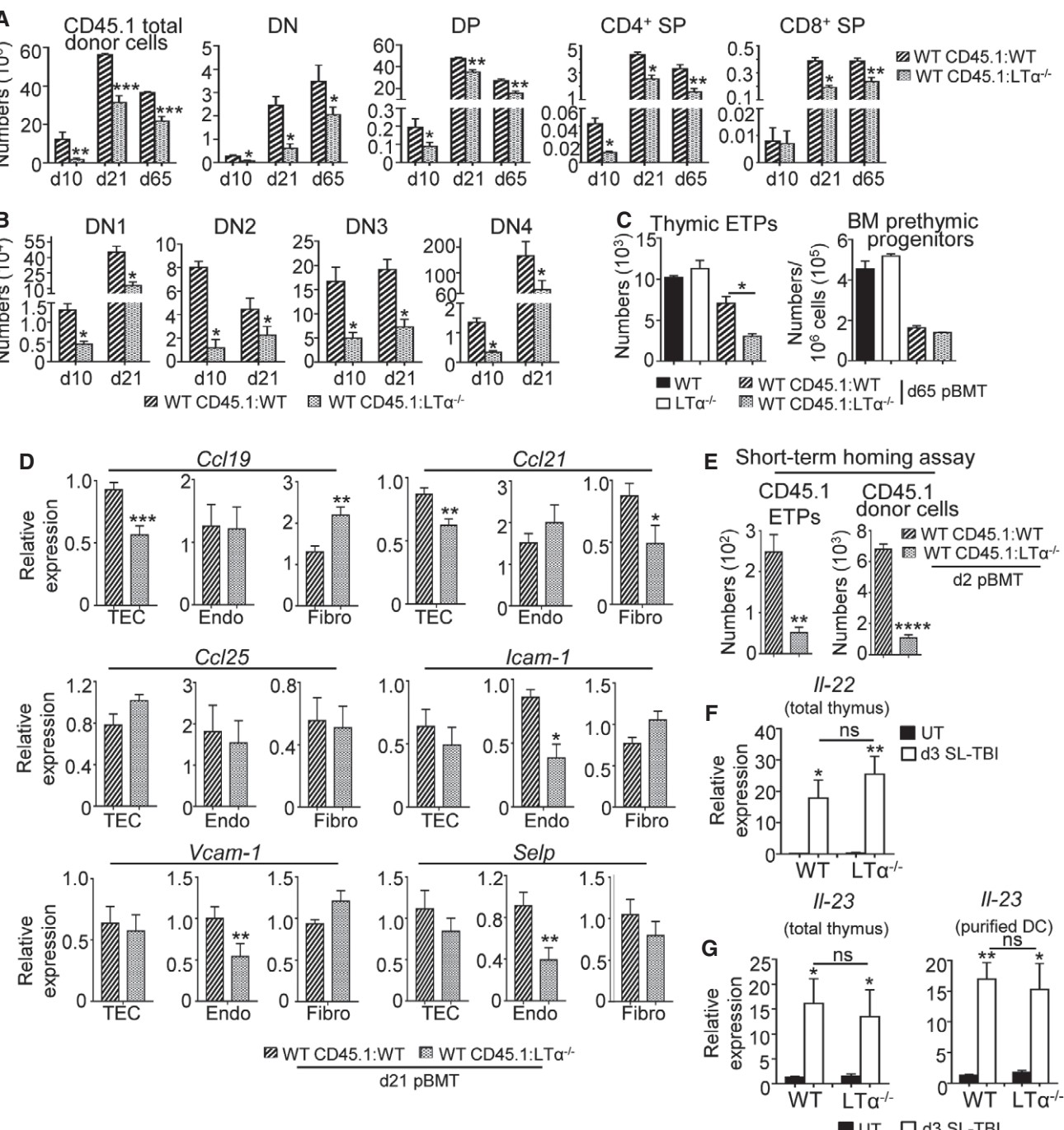

**Figure 5. LTα is required for *de novo* thymopoiesis during BMT.**

A   Numbers of total thymic cells and thymocyte subsets of CD45.1 donor origin were analyzed by flow cytometry in WT CD45.1:WT and WT CD45.1:LTα$^{-/-}$ mice at d10, d21, and d65 after BMT.

B   Numbers of DN1 (CD44$^+$CD25$^-$), DN2 (CD44$^+$CD25$^+$), DN3 (CD44$^-$CD25$^+$), and DN4 (CD44$^-$CD25$^-$) of CD45.1 origin were also analyzed at d10 and d21 after BMT.

C   Numbers of early thymic progenitors (ETPs; CD4$^-$CD8$^-$CD44$^+$CD25$^-$Lin$^-$CD117$^+$) and BM prethymic progenitors (CD3$^-$CD44$^+$CD25$^-$Lin$^-$CD117$^+$) were analyzed by flow cytometry in UT WT and LTα$^{-/-}$ mice or WT CD45.1:WT and WT CD45.1:LTα$^{-/-}$ chimeras at d65 after BMT.

D   The expression of *Ccl19, Ccl21, Ccl25, Icam-1, Vcam-1, and Selp* mRNAs was measured by qPCR in purified EpCAM$^+$ TECs, CD31$^+$EpCAM$^-$ endothelial cells, and gp38$^+$EpCAM$^-$ fibroblasts isolated from WT CD45.1:WT (*n* = 5) and WT CD45.1:LTα$^{-/-}$ (*n* = 5) chimeras at d21 after BMT.

E   Numbers of ETPs and total cells of CD45.1 donor origin in CD45.2 WT or LTα$^{-/-}$ recipients 48 h after i.v. injection of CD45.1 BM cells.

F   Expression of *Il-22* mRNA in the total thymus isolated from UT WT and LTα$^{-/-}$ mice (*n* = 4) or at d3 SL-TBI (*n* = 4).

G   Expression of *Il-23* mRNA in the total thymus and purified DCs isolated from UT WT and LTα$^{-/-}$ mice (*n* = 4) or at d3 SL-TBI (*n* = 4).

Data information: Data are shown as mean ± SEM and are pooled of three independent experiments with similar results (*n* = 3–5 mice per group). *P < 0.05; **P < 0.01; ***P < 0.001, ****P < 0.0001. Exact *P*-values and statistical tests used to calculate them are provided in Appendix Table S2.

    

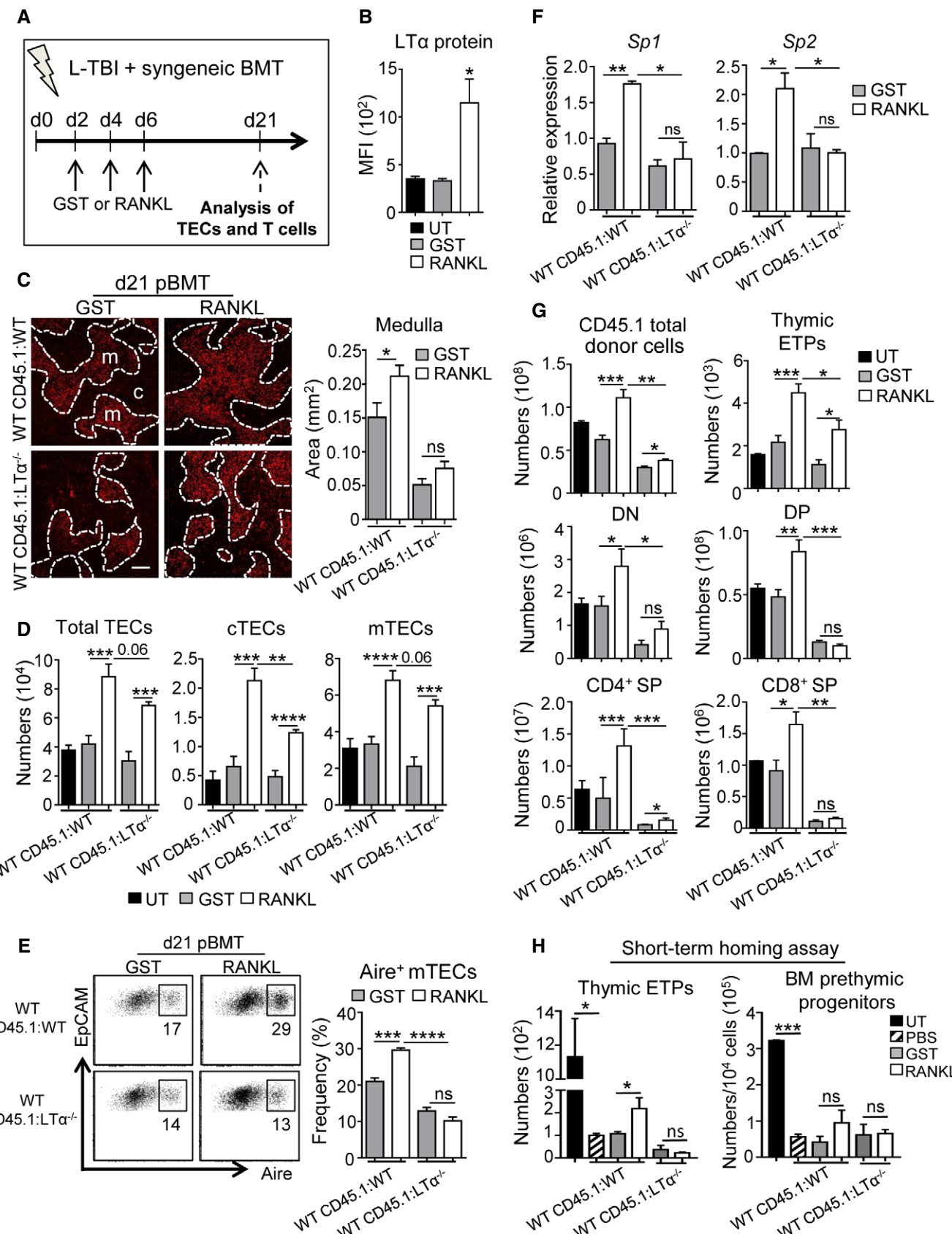

**Figure 6.**

◄

**Figure 6.  RANKL boosts TEC regeneration and *de novo* thymopoiesis in an LTα-dependent manner upon BMT.**

A    Experimental setup: WT CD45.1:WT and WT CD45.1:LTα$^{-/-}$ chimeras were treated with GST or RANKL-GST proteins at d2, d4, and d6 after BMT and TEC regeneration and T-cell reconstitution were analyzed at d21 after BMT.

B    Expression level of LTα protein in thymic LTi cells in UT mice or treated with GST or RANKL-GST.

C    Thymic sections from WT CD45.1:WT and WT CD45.1:LTα$^{-/-}$ mice treated with GST and RANKL at d2, d4, and d6 after BMT were stained for the expression of K14 at d21 pBMT. The histogram shows quantifications of medullary areas. m and c denote the medulla and the cortex, respectively. Twenty sections were quantified for each condition; scale bar: 100 μm.

D, E  Numbers of total TECs, cTECs, and mTECs (D) and flow cytometry profiles of Aire$^+$ mTECs in total EpCAM$^+$ TECs (E).

F    Expression of mRNAs coding for TRAs (*Sp1* and *Sp2*) in CD45–thymic stromal cells analyzed by qPCR.

G    Numbers of total cells and thymocyte subsets of CD45.1 donor origin analyzed in the thymus.

H    Numbers of ETPs of CD45.1 donor origin in the thymus and prethymic progenitors in the BM from CD45.2 WT or LTα$^{-/-}$ recipients 48 h after i.v. injection of CD45.1 BM cells.

Data information: Data are shown as mean ± SEM and are pooled of three independent experiments with similar results (*n* = 3–5 mice per group). **P* < 0.05; ***P* < 0.01; ****P* < 0.001, *****P* < 0.0001; Student's *t*-test. Exact *P*-values are provided in Appendix Table S2.

upregulated RANKL after irradiation, radio-resistant CD4$^+$ SP cells that expressed lower levels of RANKL constitute the major source of RANKL after TBI, which is in line with their high numbers after SL-TBI. Based on the use of a neutralizing anti-RANKL antibody and *ex vivo* RANKL administration, our data reveal that this cytokine induces LTα upregulation specifically in LTi cells after irradiation. We thus propose a model in which the effects on thymic regeneration of RANKL-regulated LTα expressed by LTi cells are

strengthened by CD4$^+$ thymocytes *via* RANKL upregulation after irradiation (Appendix Fig S8). Since RANKL, LTα, and RORγt transcription factor were upregulated in thymic LTi cells after TBI, we believe that these cells change their phenotype upon stress-induced thymic damage. Thymic LTi cells are thus likely in a "quiescent stage" at steady state and are activated after irradiation to repair the injured tissue. When considering that LTi cells expressing RANKL and LTα are involved in the organogenesis of lymph nodes (Yoshida

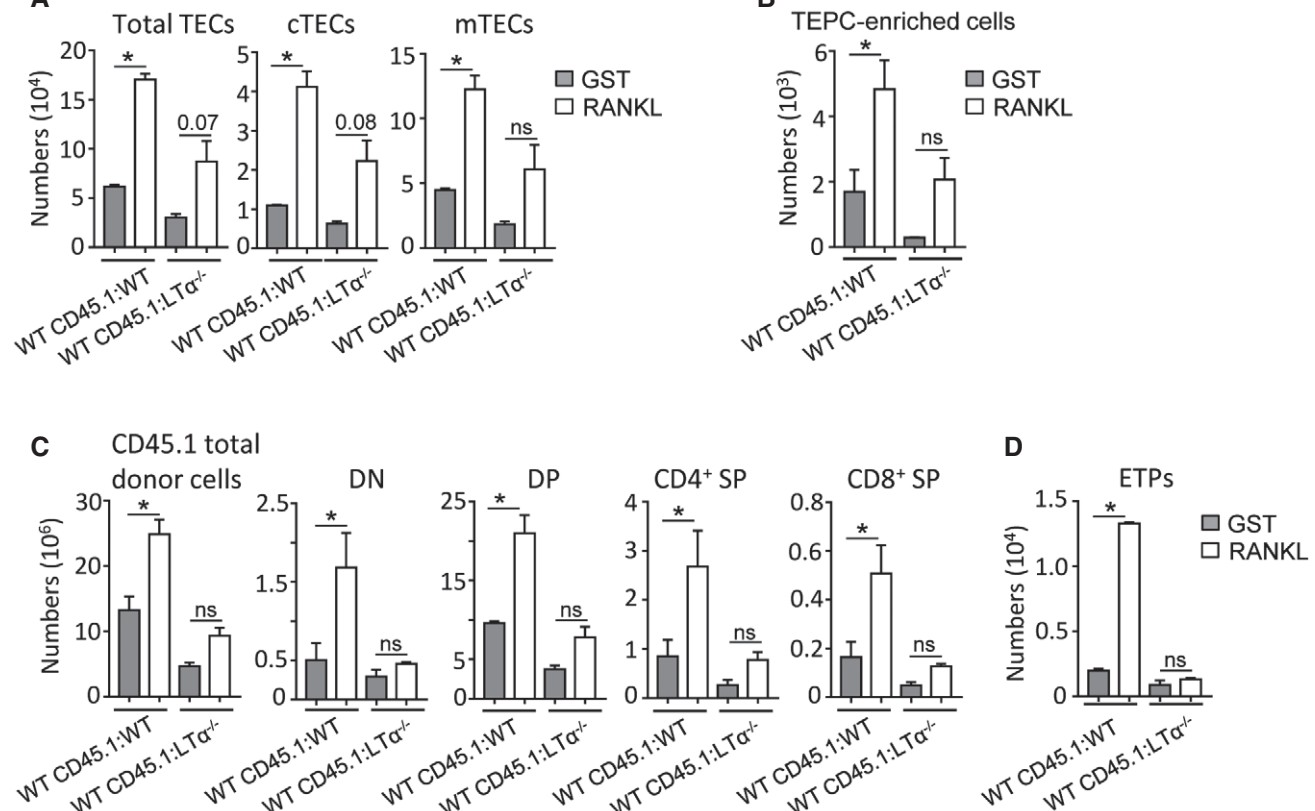

**Figure 7.  Beneficial effects mediated by RANKL treatment on thymic regeneration after BMT require LTα expression in aged mice.**

A, B  Numbers of total TECs, cTECs, mTECs (A), and TEPC-enriched cells (B) were analyzed at d21 upon BMT in the thymus from WT CD45.1:WT and WT CD45.1:LTα$^{-/-}$ chimeras of 6–8 months of age treated with GST or RANKL proteins.

C, D  Numbers of total thymic cells, thymocyte subsets (C), and ETPs (D) of CD45.1 origin.

Data information: Data are shown as mean ± SEM and are pooled of two independent experiments with similar results (*n* = 3 mice per group). **P* < 0.05; one-tailed Mann–Whitney *U*-test. Exact *P*-values are provided in Appendix Table S2.

*et al*, 1999), our data suggest that thymic LTi cells likely reactivate an embryonic program to repair the thymus after irradiation.

Interestingly, we found that the administration of a neutralizing anti-RANKL antibody led to an impaired TEC regeneration, highlighting the importance of RANKL in endogenous TEC recovery. Conversely, *ex vivo* RANKL administration after TBI boosted the regeneration of TEC subsets including TEPC-enriched cells. Furthermore, flow cytometry, histology, and qPCR experiments indicated that RANKL also boosted the regeneration of Aire$^+$ mTECs. Importantly, we demonstrate that RANKL treatment during the early phase of BMT enhanced numbers of TECs and thymocytes including ETPs as well as thymus homing of T-cell progenitors and T-cell output. Notably, although mice were treated during the early phase of BMT, RANKL treatment had long-term beneficial effects detectable until 2–3 months after BMT on both TEC and T-cell compartments. Improved T-cell reconstitution can be explained by increased stromal niches linked to increased TEC cellularity but also to enhanced thymus homing of lymphoid progenitors, which is a critical step for ameliorating T-cell recovery (Penit & Ezine, 1989; Chen *et al*, 2004; Zlotoff *et al*, 2011). The latter effect is likely mediated by increased expression of adhesion molecules and chemokines involved in this process. Nevertheless, we cannot exclude that the enhanced thymus homing in RANKL-treated mice is also favored by increased vasculature, which is important for T-cell progenitor colonization (Lind *et al*, 2001). Furthermore, given that RANKL administration upon BMT increased primitive progenitors in the BM, which correlated with the development of active osteoclasts (Appendix Fig S9; Kollet *et al*, 2006), we thus cannot also exclude that *de novo* thymopoiesis is also favored by increased hematopoietic progenitors. Whereas RANKL induces mTEC differentiation in the steady-state thymus (Rossi *et al*, 2007; Akiyama *et al*, 2008; Hikosaka *et al*, 2008; Ohigashi *et al*, 2011; Roberts *et al*, 2012), our data reveal that RANKL plays distinct roles in thymic regeneration during BMT.

We further found that RANKL induced LTα upregulation specifically in LTi cells, which express its cognate receptor, RANK. In contrast, LTα did not regulate RANKL, indicating that LTα acts downstream of RANKL. Whereas LTα is dispensable at steady state for TEC and T-cell cellularity, we found that LTα is critical for the recovery of thymic function. TEC subsets including TEPC-enriched cells were severely reduced in LTα$^{-/-}$-transplanted recipients from up to 2 months after BMT. Moreover, all thymocyte subsets as well as ETPs were also reduced in these mice likely due to defective thymus homing capacity. Consistently, we found that LTα expression is required for the expression of adhesion molecules and chemokines, known to promote T-cell progenitor entry in the thymus, in stromal cells during BMT. In accordance with our data, it has been recently reported that LTβR regulates at steady state thymus homing by controlling VCAM-1 and ICAM-1 on endothelial cells (Lucas *et al*, 2016; Shi *et al*, 2016). Peripheral T-cell reconstitution is thus altered in BM-transplanted LTα$^{-/-}$ mice, which is explained by a diminished thymic export of CD4$^+$ and CD8$^+$ T cells revealed by a weak frequency of recent thymic emigrants. Whereas LTα is involved in the regular thymic architecture at steady state (Boehm *et al*, 2003; Seach *et al*, 2008; Irla *et al*, 2012, 2013), our data therefore reveal a new function for LTα in controlling TEC and T-cell regeneration during BMT.

It has been described that the IL-10 family cytokine, IL-22, participates to thymus recovery (Dudakov *et al*, 2012). Our results show

that RANKL-regulated LTα represents a distinct pathway of that mediated by IL-22 in thymic regeneration. Furthermore, since IL-22 and LTα are both provided by LTi cells, it is likely that this cell type uses different mechanisms for thymic repair. Surprisingly, the concomitant administration of IL-22 and RANKL cytokines did not show any benefits on thymic recovery likely due to a too strong signal that results in an ineffective action on thymic recovery. Alternatively, this could be explained by a neutralization of the cell signals mediated by these two cytokines. However, while IL-22 and RANKL administrated alone show similar benefits on T-cell reconstitution, they do not exhibit the same effects on TEC regeneration with a preferential effect for IL-22 on mTEC$^{hi}$ and a more global effect for RANKL on all TEC subsets. These data thus indicate that IL-22 and RANKL play distinct roles in TEC recovery. Furthermore, one would expect that thymic regeneration during the course of BMT is also defective in RANKL$^{-/-}$ mice. Unfortunately, we were unable to test this hypothesis since RANKL$^{-/-}$ mice show severe growth retardations and exhibit a drastic reduction in mTECs (Kong *et al*, 1999; Hikosaka *et al*, 2008). Nevertheless, we demonstrate that *in vivo* RANKL neutralization impairs TEC regeneration, whereas RANKL treatment has beneficial effects on thymic recovery. Moreover, RANKL administration in BM-transplanted LTα$^{-/-}$ mice had only minor effects on TEC regeneration, numbers of ETPs and strikingly *de novo* thymopoiesis was not ameliorated. Interestingly, whereas LTα is dispensable for Aire$^+$ mTEC differentiation at steady state (Venanzi *et al*, 2007), the regeneration of these cells induced by RANKL treatment critically depends on LTα. These data indicate that the mechanisms involved in Aire$^+$ mTEC regeneration are thus distinguishable from those implicated in their emergence/differentiation at steady state. Importantly, beneficial effects of RANKL treatment in TEC regeneration are likely not due to a direct action on TECs because the effects of RANKL on these cells were modest in absence of LTα. Thus, these data argue in favor of model in which RANKL acts indirectly on TECs through LTα overexpression in LTi cells. This notion is supported by the fact that the turnover rate of mTECs is of around 2 weeks (Gabler *et al*, 2007; Gray *et al*, 2007), and thus, it is unlikely that RANKL injected early after BMT still acts on mTECs 2 months later.

Interestingly, RANKL administration is also efficient for TEC and T-cell regeneration in older individuals in which thymic involution results in diminished TEC cellularity, disrupted thymic architecture and decreased T-cell output (Gray *et al*, 2006; Chinn *et al*, 2012). These results are of special interest for elderly patients in which the recovery of T-cell functions upon BMT is less efficient (Toubert *et al*, 2012). This study thus reveals that the administration of RANKL offers an innovative clinical strategy to boost thymic recovery in young and aged individuals at several levels: TEC regeneration, thymus homing of T-cell progenitors, and *de novo* thymopoiesis. To avoid any potential side effects of the systemic administration of RANKL, such as osteoporosis, possible strategies would be to deliver directly this molecule intrathymically in patients after cytoablative conditioning or to combine RANKL with bisphosphonates to prevent bone resorption (Tomimori *et al*, 2009). The administration of bisphosphonates induces apoptosis of mature osteoclasts (Hughes *et al*, 1995; Drake *et al*, 2008) and importantly does not affect RANKL synthesis (Kim *et al*, 2002; Verde *et al*, 2015). In conclusion, RANKL in clinic is expected to be promising for enhancing the regeneration of immune functions in patients whose thymus has been severely damaged.

# Materials and Methods

### Mice

CD45.1 and CD45.2 WT (Janvier), CD45.2 LTα$^{-/-}$ (De Togni et al, 1994), Rorc$^{GFP/GFP}$ knock-in (Eberl et al, 2004), ZAP-70$^{-/-}$ (Wiest et al, 1997), and Rag2$^{-/-}$ (Shinkai et al, 1992) mice were on B6 background and maintained under specific pathogen-free conditions at the CIML, AniCan (Lyon, France) or IGMM (Montpellier, France). Standard food and water were given ad libitum. Males and females were used at the age of 6–8 weeks for each mouse strain. Chimeras were generated at 6–8 weeks or 6–8 months of age.

### Thymic damage and BM chimeras

Total body irradiation was performed with a Cs-137 γ-radiation source. Sublethal-TBI (SL-TBI) was performed with 500 rads with no hematopoietic rescue and L-TBI with 2 doses of 500 rads. Dexamethasone (20 mg/kg, Sigma-Aldrich) treatment was performed by intraperitoneal injection. For the generation of chimeras, $10^7$ BM cells of CD45.1 origin were injected i.v. into lethally irradiated (2 × 500 rads) CD45.2 WT or LTα$^{-/-}$ recipient mice.

### RANKL and IL-22 stimulations

The recombinant mouse RANKL-GST protein was produced as previously described (Knoop et al, 2009). RANKL-GST (5 mg/kg) or GST (5 mg/kg) proteins were administrated i.v. daily during 3 days after SL-TBI or at days 2, 4, and 6 after BMT. Recombinant mouse IL-22 protein (200 μg/kg; R&D Systems) was administrated i.v. at days 2, 4, and 6 after BMT in combination or not with RANKL-GST protein.

### RANKL neutralization experiments

150 μg of Low Endotoxin and Azide-Free (LEAF) neutralizing anti-RANKL antibody (IK22/5; BioLegend) or purified Rat IgG2a, κ isotype control (RTK2758; BioLegend) were administrated i.v. during 3 days after SL-TBI.

### Stromal cell isolation

Stromal cells were isolated as previously described (Irla et al, 2008) by enzymatic digestion with collagenase D and DNase I (Roche) and depletion of hematopoietic cells using anti-CD45 magnetic beads and AutoMACS (Miltenyi Biotec). TECs, endothelial cells, and fibroblasts were cell-sorted with EpCAM, CD31, and gp38 markers, respectively, with a FACSAriaIII cell sorter (BD).

### Flow cytometry

CD4 (1:600; RM4.5), CD8α (1:600; 53-6.7), CD45.1 (1:800; A20), LTα (1:80; AF.B3), IL-7Rα (1:80; SB/199), CD80 (1:200; 16-10A1), Ly51 (1:3,000; BP-1), I-Ab (1:200; AF6-120.1), CD45 (1:200; 30-F11), CD44 (1:200; IM7), CD62L (1:300; MEL-14), and Sca-1 (1:500; D7) antibodies were from BD. CD25 (1:200; PC61), RANK (1:200; R12-31), RANKL (1:200; IK22/5), CD3ε (1:200; 145-2C11), lineage cocktail (20 μl/million cells; 145-2C11, RB6-8C5, M1/70, RA3-6B2, Ter-119), CD11c (1:200; N418), α6-integrin (1:200; GoH3), and

CD31 (1:200; 390) were from BioLegend. Foxp3 (1:200; FJK-16s), Ki-67 (1:300; SolA15), EpCAM (1:3000; G8.8), LTβR (1:200; ebio3C8), Aire (1:200; 5H12), RORγt (1:300; B2D), CD117 (1:200; 2B8), and PDGFRα (1:200; APA5) were from eBioscience. FITC-conjugated UEA-1 lectin (1:800) was from Vector Laboratories. For LTα detection, cells were incubated for 3 h with brefeldin A (Biosciences). Foxp3 and Ki-67 intracellular stainings were performed with the Foxp3 staining kit (eBioscience). Aire, LTα, RANKL, and RORγt intracellular stainings were performed with BD Cytofix/ Cytoperm and Perm/Wash buffers. For staining with LTβR-Fc, cells were incubated with LTβR-Fc (RnD systems) at 1 μg/$10^6$ cells for 45 min on ice. LTβR-Fc staining was visualized using an Alexa Fluor 488-conjugated donkey anti-human IgG F(ab')$_2$ fragment (1:400; Jackson ImmunoResearch). Flow cytometry analysis was performed with a FACSCanto II (BD), and data were analyzed with FlowJo software.

### Quantitative RT–PCR

Total RNA was prepared with TRIzol (Invitrogen). cDNAs were synthesized with oligo(dT) using Superscript II reverse transcriptase (Invitrogen). qPCR was performed with the ABI 7500 fast real-time PCR system (Applied Biosystem) and SYBR Premix Ex Taq master mix (Takara). Primers are listed in Appendix Table S1.

### Signal joint TREC analysis

Signal joint TREC analysis to detect recent thymic emigrants was performed as described previously (Sempowski et al, 2002). Briefly, genomic DNA was isolated from purified CD4$^+$ and CD8$^+$ splenic T cells using the QIAamp DNA Mini Kit (Qiagen). Real-time PCR quantification of sjTRECs was performed using the CD45 reference gene to correct genomic DNA input.

### Immunofluorescence staining

Frozen thymic sections were stained with Alexa Fluor 488-conjugated anti-Aire (1:200; 5H12, ebioscience) and anti-keratin 14 (1:800; AF64, Covance Research) revealed with Cy3-conjugated anti-rabbit (1:500; Invitrogen) and counterstained with 1 μg/ml DAPI as previously described (Serge et al, 2015). Images were acquired with a LSM 780 Leica SP5X confocal microscope and quantified with ImageJ software.

### TRAP staining

Mouse femurs were fixed in 4% paraformaldehyde during 48 h and were decalcified with 10% EDTA, pH 7.5 during 15 days. Deparaffinized 5-μm sections were stained for TRAP activity (Sigma) according to the manufacturer's instructions. Sections were counterstained with hematoxylin, and images were quantified with ImageJ software.

### Statistical analysis

Statistical significance was assessed with GraphPad Prism 6 software using unpaired Student's t-test, Mann–Whitney test, or ANOVA on multiple variable analyses *$P < 0.05$; **$P < 0.01$;

## The paper explained

### Problem

Cytoablative treatments commonly used to prepare patients to bone marrow transplantation (BMT) severely affect thymic stromal cells, which results in delayed T-cell reconstitution. This prolonged deficiency in T cells could result in severe clinical consequences characterized by an increased susceptibility to opportunistic infections, autoimmunity, tumor relapse, or the development of secondary malignancies and thus could lead to post-transplant morbidity and mortality. Although the mechanisms that govern thymic regeneration are crucial for the recovery of a functional immune system, they remain poorly understood. Furthermore, the identification of therapeutic molecules that boost thymic regeneration is of clinical interest to efficiently improve T-cell reconstitution and thus to prevent a period of compromised immunity after cytoablative treatments.

### Results

Here, we found that RANKL, a TNF family member, is naturally upregulated in CD4$^+$ thymocytes and lymphoid tissue inducer (LTi) cells during the early phase of thymic recovery. Importantly, the *in vivo* neutralization of RANKL impairs thymic epithelial cell (TEC) regeneration, whereas the *ex vivo* administration of RANKL protein boosts thymic recovery during the course of BMT. We further demonstrate that RANKL treatment after total body irradiation induces specifically in LTi cells, LTα expression, which is critical for the regeneration of TECs and *de novo* thymopoiesis. Importantly, thymic recovery improved by RANKL administration strictly depends on LTα expression. Furthermore, RANKL treatment boosts thymic recovery upon BMT not only in young but also in aged individuals.

### Impact

In the present work, we identified RANKL as a new potential candidate for boosting thymic regeneration after thymic damage. This finding opens a new perspective for ameliorating the recovery of T-cell functions not only after cytoablative therapy but also in patients with complications related to aging process, HIV/AIDS, malnutrition, or radiation poisoning due to nuclear disaster.

***$P < 0.001$, ****$P < 0.0001$. Normal distribution of the data was assessed using d'Agostino-Pearson omnibus normality test. Correlations were calculated using the nonparametric Spearman correlation test. Error bars represent mean ± SEM. Exact *P*-value and statistical test used for each figure are provided in Appendix Table S2.

### Study approval

Experiments were performed in accordance with the animal care guidelines of the European Union and French laws. All animal procedures were approved by and performed in accordance with guidelines of the Centre d'Immunologie de Marseille-Luminy (CIML) and AniCan (Lyon, France).

**Expanded View** for this article is available online.

### Acknowledgements

We thank P. Ferrier (CIML, France) for his support, G. Hollander (Basel, Switzerland) and A. Sergé (CRCM, France) for critical reading of the manuscript, C. Fauriat (CRCM, France) for discussions, L. Chasson and J. Charaix (CIML, France) for help in histology. We are grateful to N. Taylor and V. Zimmermann (IGMM, Montpellier) for providing us thymus from Zap-70$^{-/-}$ mice, IR. Williams (Atlanta, USA) and C. Mueller (IBMC, France) for providing us the RANKL-GST construct and J.H Veiga Fernandes (Lisboa, Portugal) for providing Rorc$^{GFP/GFP}$ knock-in mice. We acknowledge the flow cytometry and animal facility platforms at CIML for excellent technical support. This work was supported by institutional grants from Institut National de la Santé et de la Recherche Médicale, Centre National de la Recherche Scientifique and Aix-Marseille Université, the Swiss National Science Foundation (AmbizionePZ00P3_131945 to M.I.), the Marie Curie Actions (Career Integration Grants, CIG_SIGnEPI4Tol_618541 to M.I.), the Jules Thorn Foundation (to M.I.), and Helmholtz-DKFZ (to J.M.). We acknowledge financial support from no. ANR-10-INBS-04-01 France Bio Imaging. N.L. was supported by a doctoral fellowship from Aix-Marseille University.

## Author contributions

NL, HV, JM, and MI performed experiments and analyzed the data. NL and MI interpreted data and wrote the manuscript. MI supervised and conceived the study.

## Conflict of interest

The authors declare that they have no conflict of interest.

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
