## [Review Process File · EMBO Molecular Medicine]

Administration of RANKL boosts thymic regeneration upon bone marrow transplantation

Noëlla Lopes, Hortense Vachon, Julien Marie and Magali Irla

Corresponding author: Magali Irla, Centre d'Immunologie de Marseille-Luminy CIML

Review timeline:

Submission date:	14 October 2016
Editorial Decision:	17 November 2016
Revision received:	28 February 2017
Editorial Decision:	07 March 2017
Revision received:	22 March 2017
Accepted:	24 March 2017

Transaction Report:

Editor: Céline Carret

1st Editorial Decision

17 November 2016

Thank you for the submission of your manuscript to EMBO Molecular Medicine. We have now heard back from the two referees whom we asked to evaluate your manuscript. As you will see from the reports below, the referees find the topic of your study of potential clinical interest. However, they raise substantial concerns on your work, which should be convincingly addressed in a major revision of the present manuscript.

Particularly, both referees find the study insufficient to make a compelling case and while referee 1 is more enthusiastic, this referee would be interested in a better description of RANKL and LT α production after TBI in the thymus and by the demonstration of the relationship with the IL22/23 pathway of thymic protection. As for referee 2, this reviewer is not convinced by the cellular source of RANKL and several downstream interpretations of the data. Additional *in vivo* experiments are needed to address all these issues.

Given that they nevertheless find the message of the study novel and interesting, we would be willing to consider a revised manuscript with the understanding that the referees concerns must be fully addressed and that acceptance of the manuscript would entail a second round of review.

Overall it is clear that publication of the manuscript cannot be considered at this stage. I also note that addressing the reviewers concerns in full will be necessary for further considering the manuscript in our journal and this appears to require a lot of additional work and experimentation. I am unsure whether you will be able or willing to address those and return a revised manuscript within the 3 months deadline. On the other hand, given the potential interest of the findings, I would be willing to consider a revised manuscript with the understanding that the referees' concerns must be fully addressed and that acceptance of the manuscript would entail a second round of review.

I should remind you that it is EMBO Molecular Medicine policy to allow a single round of revision only and that, therefore, acceptance or rejection of the manuscript will depend on the completeness of your responses included in the next, final version of the manuscript. For this reason, and to save you from any frustrations in the end I would strongly advise against returning an incomplete revision and would also understand your decision if you chose to rather seek rapid publication elsewhere at this stage.

Should you decide to revise for EMBO Molecular Medicine though, revised manuscripts should be submitted within three months of a request for revision; they will otherwise be treated as new submissions, except under exceptional circumstances in which a short extension is obtained from the editor.

I look forward to seeing a revised form of your manuscript.

Should you find that the requested revisions are not feasible within the constraints outlined here and choose, therefore, to submit your paper elsewhere, we would welcome a message to this effect.

***** Reviewer's comments *****

Referee #1 (Remarks):

In their manuscript, Lopes and al, addressed the problem of immune reconstitution after thymic damage which is medically in situation such as Bone Marrow Transplantation (BMT). They showed that injection of recombinant RANKL protein was able to improve thymic reconstitution through the production of LTalpha by LT_i cells after an autocrine RANK RANKL loop. Production of LTalpha by LT_i induced regeneration of almost all Thymic Epithelial cells favoring Thymus seeding by T lymphocyte progenitors and thymocyte and lymphocyte production. For this they used a mouse model of BMT with TBI and LTalpha KO mouse. Finally they showed that LT_i mediated RANKL action was still efficient in older mouse with lower thymic function.

Major Remarks

This a well done study with a lot of work. This is probably why the authors did not push a little bit further the investigation. I would be very interested to know the effect of RANKL in normal mice and especially if RANKL could reverse the effect of aging on thymic involution. In the same way, if the authors showed that RANKL effect was different of IL23/IL22 thymic protection that have the same target cells (ie LT_i) because it does not induce these cytokines expression, it would be very interesting to me to combined these treatment in order to see if these effects could be additive, synergic or use the same downstream pathway.

Authors claimed that LT_i were the main source of RNKL and LTalpha, they should take into account the low number of LT_i cells (less than .1 %) compared to the huge amount of DP

Minor remarks

Data on naïve T cell peripheral reconstitution (and even better on TREC reconstitution) would be better than whole CD4 or CD8 number since it's less subject to homeostatic proliferation. Fluorescence scale is missing on Flow cytometry dot plot which does not allow us to compare RANKL an LT alpha expression on the different cell subtype.

Fig 2B Should the third histogram not be labeled cTEC_{low} ?

Sup Figure 2 E : is the legend right ?

Referee #2 (Remarks):

This study reports using experimental mouse models that RANKL is strongly up-regulated in the thymus during the thymic regeneration caused by total body irradiation. The authors also found that administration of RANKL protein after irradiation and bone marrow transplantation promoted the regeneration of thymic stromal cells including TEC subsets, and enhanced the T-progenitor homing and thymic T cell development. They show some data that RANKL administration increases the

expression of LTa in LTi cells and LTa is critical for TEC regeneration and T cell reconstitution after bone marrow transplantation, and then propose a model that thymic LTi cells are the key player in the RANKL-mediated thymic regeneration.

The authors are correct to state that administration of RANKL boosts thymic regeneration upon irradiation and bone marrow transplantation in mouse. So, this study may lead to the development of clinical strategies to boost thymic recovery upon bone marrow transplantation. However, given that RANKL is an activator of osteoclastogenesis and osteoclast activity, the administration of huge amount of RANKL must be unrealistic in clinical situations, even though the authors discuss the possibility of intrathymic delivery of RANKL and prevention of bone resorption by combined bisphosphonate treatment. Furthermore, there are also serious concerns about the mechanisms that the authors propose for the function of RANKL, which I believe preclude the publication of this ms in its current form.

First, the authors stated that LTi cells are the main producers of RANKL in irradiated thymus. Their flow cytometry data clearly show that the expression level of RANKL after irradiation is highest in LTi cells among thymic lymphoid cells (Fig. 1B). However, the contribution of LTi cells to thymic RANKL production might be unclear, because the frequency of LTi cells in the thymus is very small (~0.1% in irradiated thymus, as shown in Suppl. Fig. 2E). Rather, CD4SP cells, another major producer of RANKL, are the most prominent lymphoid cell population (~50%) in the irradiated thymus, as reported previously (J. Exp. Med 200, 493-505, 2004). Is it possible that CD4SP cells but not LTi cells are the main producers of RANKL in this experimental setting? Or, do LTi cells robustly increase in the irradiated thymus and take the place of CD4SP cells? To make this issue clear, the flow cytometry profiles and cell number data of LTi cells and other lymphoid cells including CD4SP cells before and after irradiation must be shown.

The authors show that the RANKL expression after irradiation is significantly lower in *Rorc*^{-/-} thymus than in WT thymus (Fig. 1D), to claim that LTi cells are critical for RANKL up-regulation after thymic damage. However, this interpretation might be incorrect, because *Rorc*^{-/-} mice also exhibit severely reduced CD4SP cells (Science 288, 2369-2373, 2000). Thus, in addition to *Rorc*^{-/-} mice, *Tera*^{-/-} mice should be used; *Tera*^{-/-} mice have normal number of LTi cells but completely lack CD4SP cells, so if LTi cells were the main source of RANKL, they would show the high RANKL expression similar to WT.

The authors also concluded that LTi cells are the main producer of LTa in irradiated thymus (Fig. 3), but this interpretation is also questionable, because LTa is also highly expressed in CD4SP cells.

Second, the authors show that thymic LTi cells express high level of RANK after irradiation, and the RANKL-RANK signal induces in LTi cells, LTa, which is critical for thymic regeneration. In view of this, it is worth to ask whether the LTa expression and thymic regeneration are impaired in the absence of RANKL, using RANKL-flox or RANK-flox mice crossed with lymphoid-specific Cre mice. Alternatively, the loss-of-function study on RANKL can be achieved if the authors use RANK-Fc protein. Whether the administration of RANK-Fc into irradiated mice could inhibit the LTa up-regulation and thymic regeneration must be determined.

Lastly, it was also reported that RANKL stimulates osteoclasts in the bone to promote the mobilization of lymphoid progenitor cells (Nat. Med. 12, 657-664, 2006). The authors should examine and discuss this possibility.

Minor points:

1. Fig. 2H shows that the RANKL administration influences gene expression in cTECs. How can RANKL affect cTECs?

2. In Fig. 5D, the authors examined the gene expression in CD45- thymic stromal cells and normalized the expression levels with EpCAM or CD31. However, CD45- thymic stroma contain not only TECs and endothelial cells but also other stromal cells such as fibroblasts, and some genes such as CCL25, CCL19 and CCL21 are also expressed in fibroblasts (J. Immunol., 178, 4956-4965, 2007). Thus, the data in Fig. 5D do not indicate the defective gene expression in TECs and endothelial cells.

3. Why did the authors need to perform BFA treatment and intracellular staining for detecting RANKL or LT α protein expression? Are those treatments needed for LT β R-Fc staining?
4. In Fig. 2A and Fig. 4B, flow cytometry profiles for total TECs in total live cells are shown. It is supposed that these are not total live thymic cells but likely MACS-enriched or CD45-negative stromal cells. Must be described in the figure legends.

1st Revision - authors' response

28 February 2017

Referee #1 (Remarks):

In their manuscript, Lopes and al, addressed the problem of immune reconstitution after thymic damage which is medically in situation such as Bone Marrow Transplantation (BMT). They showed that injection of recombinant RANKL protein was able to improve thymic reconstitution through the production of LT α by LTi cells after an autocrine RANK RANKL loop. Production of LT α by LTi induced regeneration of almost all Thymic Epithelial cells favoring Thymus seeding by T lymphocyte progenitors and thymocyte and lymphocyte production. For this they used a mouse model of BMT with TBI and LT α KO mouse. Finally they showed that LTi mediated RANKL action was still efficient in older mouse with lower thymic function.

MAJOR REMARKS This a well done study with a lot of work. This is probably why the authors did not push a little bit further the investigation. I would be very interested to know the effect of RANKL in normal mice and especially if RANKL could reverse the effect of aging on thymic involution.

Response: We would like to thank the reviewer for underlining the amount of work already provided in the initial version of the manuscript and regarded the quality of our study. We agree with the reviewer that the role of RANKL in “normal mice“ and during aging is of particular interest. Besides the role of *ex vivo* administration of RANKL protein on thymic regeneration upon bone marrow transplantation (BMT), we investigated the effect of this treatment on thymopoiesis in young and aged mice at steady state. Several sets of experiments indicate that RANKL administration substantially enhances TEC numbers, thymopoiesis and T-cell output both in young and aged WT mice (Manuscript in preparation). Given that age-related thymic involution is associated with reduced T-cell output and low diversity for specific antigen recognition, RANKL could thus constitute a promising therapeutic molecule to help in reversing the effects of thymic involution. However, we believe that these results constitute an independent finding from the current manuscript, which is specifically focused on thymic regeneration after irradiation during the course of BMT. Moreover, though we do not have yet the full mechanisms of RANKL action on thymic involution, our preliminary data indicate that the role of RANKL on thymic involution obeys to different mechanisms of those described after irradiation. We therefore did not include the data concerning the effects of RANKL in young and aged mice at steady state. However, they can be mentioned in the discussion as personal observation or alternatively, we will be happy to provide them at the reviewer’s discretion.

In the same way, if the authors showed that RANKL effect was different of IL23/IL22 thymic protection that have the same target cells (ie LTi) because it does not induce these cytokines expression, it would be very interesting to me to combined these treatments in order to see if these effects could be additive, synergic or use the same downstream pathway.

Response: We thank the reviewer for this interesting suggestion. In this regard, we performed several sets of experiments to analyze the effects of the concomitant administration of IL-22 and RANKL in thymic regeneration after BMT. WT mice transplanted with CD45.1 bone marrow cells were treated with IL-22, RANKL or IL-22+RANKL at day 2, 4 and 6 after BMT and thymic regeneration was analyzed at d21 after BMT. The results of these experiments are now presented in the revised version in Appendix Figure S6. Unexpectedly, the combined administration of IL-22 and RANKL did not substantially ameliorate thymic regeneration as single administration does. We believe that this result is likely due to a too strong effect (when they are administrated simultaneously) on TECs or alternatively to the neutralization of cell signals mediated by these two cytokines. These possibilities are discussed in the discussion part (cf. lines 444448). However, IL-22 and RANKL administrated alone similarly improved T-cell reconstitution (cf. Appendix Fig S6A-C). Interestingly, these experiments were extremely informative since they showed that IL-22 and

RANKL alone plays distinct roles in TEC recovery. We found that IL-22 increases preferentially numbers of mTEC^{hi} (MHCII^{hi}UEA-1⁺) cells including CD80^{hi}Aire⁻ and CD80^{hi}Aire⁺ mTEC subsets (cf. Appendix Fig S6E,F) whereas RANKL enhances all TEC populations, including cTEC and mTEC subsets (cf. Appendix Fig S6D-F). Furthermore, RANKL also increased thymic epithelial progenitor (TEPC)-enriched cells as defined in Wong *et al.* (Cell reports, 2014) (cf. Appendix Fig S6G). In our BMT experimental setting, RANKL thus seems to have a broader effect on TEC regeneration compared to IL-22. In view of these results, we thus believe that in addition to IL-22, RANKL constitutes another efficient cytokine to improve thymic regeneration upon BMT by enhancing both TEC and T-cell compartment.

Authors claimed that LTi were the main source of RANKL and LTα, they should take into account the low number of LTi cells (less than .1 %) compared to the huge amount of DP

Response: We fully agree that LTi cells represent a rare cell population in the thymus (cf. Figure 1C and Table 1). In this revised version, we examined in addition to LTi cells the number of lymphoid cell populations before and after irradiation. Dot plots and numbers are now presented in Figure 1B-C and in Table 1, respectively. At day 3 after SL-TBI, we observed that DP thymocytes are massively eliminated ($130 \times 10^6 \pm 14.4 \times 10^6$ cells in untreated mice vs $0.27 \times 10^6 \pm 0.07 \times 10^6$ after irradiation, *i.e.* a decrease of ~480 fold), as previously reported (Randle-Barrett ES and Boyd RL, Dev Immunol. 1995). In contrast, we observed that CD4⁺ thymocytes constitute the main lymphoid cell population after irradiation ($18.01 \times 10^6 \pm 2.07 \times 10^6$ in untreated mice vs $2.6 \times 10^6 \pm 0.2 \times 10^6$ after irradiation, *i.e.* a decrease of ~7 fold) in accordance with the study of Ueno *et al.* (JEM; 2004). Furthermore, consistently with Dudakov *et al.* (Science; 2012), we observed that LTi cells constitute a partially radio-resistant cell type ($3.35 \times 10^3 \pm 0.72 \times 10^3$ cells in untreated WT mice vs $1.58 \times 10^3 \pm 0.12 \times 10^3$ cells after irradiation; *i.e.* a decrease of ~2 fold).

In the original version of the manuscript, we used the LTi-deficient thymus from RORc^{-/-} mice to analyze the contribution of LTi cells in both RANKL and LTα upregulation after irradiation. However, as mentioned by Reviewer #2, RORc^{-/-} mice exhibit drastic reduced numbers of CD4⁺ thymocytes (Sun *et al.*, Science, 2000). In order to remove any concerns, we completed our investigation in this revised version by using genetically deficient mice: ZAP-70^{-/-} mice (lacking SP cells) and Rag2^{-/-} mice (lacking DP and SP cells), which both exhibit LTi cells. As now presented in Figure 1E, we found that irradiated ZAP-70^{-/-} mice failed to increase RANKL in their thymi, excluding a potential implication of DP cells and indicating that CD4⁺ thymocytes are the main source of RANKL after irradiation. We also provide flow cytometry profiles showing that CD4⁺ thymocytes upregulated RANKL during the course of SL-TBI with no hematopoietic rescue but at a lesser extent than LTi cells (Figure 1D,F). In order to further define the contribution of LTi cells in RANKL upregulation, we also analyzed the thymus of Rag2^{-/-} mice, which upregulated RANKL after irradiation but at lesser extent than in WT thymi (Figure 1E), confirming that LTi cells also contribute to RANKL overexpression after TBI.

In contrast to RANKL, using the same genetic approaches, we found that irradiated ZAP-70^{-/-} thymi upregulated LTα expression at the same level than that of irradiated WT thymi (cf. Figure 3F), indicating that although CD4⁺ thymocytes are the most representative lymphoid population after SL-TBI, they are not involved in LTα upregulation after irradiation. However, since ZAP-70^{-/-} mice have DP and LTi cells, we then investigated the role of these two cell types in LTα overexpression after irradiation. For this, we used irradiated Rag2^{-/-} mice (lacking DP but showing LTi cells), which expressed LTα at the same level than that of irradiated WT mice. This result thus confirms that LTα is largely produced by LTi cells after SL-TBI and that DP cells are not involved in this process (cf. Figure 3G). The latter result was also confirmed with irradiated WT thymus deprived three days before of DP cells by dexamethasone treatment, a procedure described in Purton, *et al.*, J Immunol. 2004 (cf. Figure 3G). These new data thus confirmed that only LTi cells are critical in LTα upregulation after irradiation.

In sum, with these genetic approaches, we provide evidences that RANKL is mainly provided by CD4⁺ thymocytes after irradiation, and that though thymic LTi cells represent a small cell subset, they contribute to RANKL upregulation and constitute the main provider of LTα upregulation after irradiation. The model recapitulating our new findings is presented in Appendix Figure S8.

MINOR REMARKS Data on naïve T cell peripheral reconstitution (and even better on TREC reconstitution) would be better than whole CD4 or CD8 number since it's less subject to homeostatic proliferation.

Response: We appreciate the reviewer suggestion. We completed our analysis in BM-transplanted WT and $LT\alpha^{-/-}$ mice by analyzing naïve peripheral T cells identified as $CD62L^{+}CD44^{+}$ by flow cytometry. We found a reduced number of both naïve $CD4^{+}$ and $CD8^{+}$ T cells in the spleen of BM-transplanted $LT\alpha^{-/-}$ mice compared to BM-transplanted WT mice. These new results are shown in Figure EV3E. Furthermore, as suggested by the reviewer, we quantified signal joint TCR excision circles (sjTRECs) by quantitative PCR and found a reduced detection of sjTRECs in peripheral T cells of $LT\alpha^{-/-}$ recipient mice. These data are shown in Figure EV3F. Altogether, these additional experiments thus firmly confirm a diminished thymic activity in BM transplanted- $LT\alpha^{-/-}$ mice.

Fluorescence scale is missing on Flow cytometry dot plot which does not allow us to compare RANKL an LT alpha expression on the different cell subtype.

Response: We are sorry for not including the fluorescence scale in our FlowJo analysis. Modifications were made in this revised version.

Fig 2B Should the third histogram not be labeled cTEC^{low}?

Response: We have used the definition of the different TEC subsets described in the study of Wong K *et al.*, *Cell Reports*. 2014. Adult TECs were divided into discrete subsets based on the expression of MHCII, UEA-1 and Ly51 markers: -TEC^{lo} (MHCII^{lo}UEA-1⁻Ly51^{lo}), -cTEC^{hi} (MHCII^{hi}UEA-1⁻Ly51^{hi}), -mTEC^{lo} (MHCII^{lo}UEA-1⁺Ly51⁻), -mTEC^{hi} (MHCII^{hi}UEA-1⁺Ly51⁻). The third histogram was thus properly labeled “TEC^{low}”.

Sup Figure 2 E : is the legend right ?

Response: We are sorry for the confusion. In this revised version, we have corrected this mistake by indicating in the panel E “Gated on $CD4^{+}CD8^{-}$ cells”.

Referee #2 (Remarks):

This study reports using experimental mouse models that RANKL is strongly up-regulated in the thymus during the thymic regeneration caused by total body irradiation. The authors also found that administration of RANKL protein after irradiation and bone marrow transplantation promoted the regeneration of thymic stromal cells including TEC subsets, and enhanced the T-progenitor homing and thymic T cell development. They show some data that RANKL administration increases the expression of $LT\alpha$ in LTi cells and $LT\alpha$ is critical for TEC regeneration and T cell reconstitution after bone marrow transplantation, and then propose a model that thymic LTi cells are the key player in the RANKL-mediated thymic regeneration. The authors are correct to state that administration of RANKL boosts thymic regeneration upon irradiation and bone marrow transplantation in mouse. So, this study may lead to the development of clinical strategies to boost thymic recovery upon bone marrow transplantation.

However, given that RANKL is an activator of osteoclastogenesis and osteoclast activity, the administration of huge amount of RANKL must be unrealistic in clinical situations, even though the authors discuss the possibility of intrathymic delivery of RANKL and prevention of bone resorption by combined bisphosphonate treatment. Furthermore, there are also serious concerns about the mechanisms that the authors propose for the function of RANKL, which I believe preclude the publication of this ms in its current form.

Response: We would like to thank the reviewer for her/his constructive comments that have substantially strengthened our revised version. We totally agree with the reviewer that RANKL is an activator of osteoclastogenesis. However, since several studies both in mouse and human have shown that bisphosphonates are potent inhibitors of bone resorption (Hughes DE *et al.*, *J Clin Invest*, 1989; Hughes DE *et al.*, *J Bone Miner Res*, 1995; Parfitt AM *et al.*, *J Bone Miner Res*, 1996; Jilka RL *et al.*, *J Clin Invest*, 1998) and are the most widely used treatment for osteoporosis

(Wysowski DK and Greene P, Bone, 2013), we believe that RANKL could be administered in combination with bisphosphonates specifically because:

- (1) they decrease the differentiation of osteoclasts and induce their apoptosis (Drake MT *et al.*, Mayo Clin Proc, 2008).
- (2) in contrast to denosumab (a neutralizing anti-RANKL antibody) and Xanthohumol (that disrupts the association between RANK and TRAF6), bisphosphonates do not interfere with RANKL to inhibit bone resorption (Cummings SR *et al.*, N Engl J Med, 2009; Li J *et al.*, Sci Rep, 2015; Verde M *et al.*, Acta Oncol.Latinoam, 2015; Kim YH *et al.*, Exp Mol Med 2002).
- (3) combined with RANKL treatment, they suppress both mouse and human osteoclast differentiation mediated by RANKL (Tomimori *et al.*, J Bone Miner Res, 2009).

Alternatively, RANKL could be also delivered locally in the thymus to avoid any potential effects on the bone. In line with this possibility, the feasibility of intrathymic injections has been already successfully realized in macaques under endoscopic guidance (Moreau A *et al.*, Mol Ther. 2009 and De Barros *et al.*, Stem Cells, 2013).

First, the authors stated that LTi cells are the main producers of RANKL in irradiated thymus. Their flow cytometry data clearly show that the expression level of RANKL after irradiation is highest in LTi cells among thymic lymphoid cells (Fig. 1B). However, the contribution of LTi cells to thymic RANKL production might be unclear, because the frequency of LTi cells in the thymus is very small (~0.1% in irradiated thymus, as shown in Suppl. Fig. 2E). Rather, CD4SP cells, another major producer of RANKL, are the most prominent lymphoid cell population (~50%) in the irradiated thymus, as reported previously (J. Exp. Med 200, 493-505, 2004). Is it possible that CD4SP cells but not LTi cells are the main producers of RANKL in this experimental setting? Or, do LTi cells robustly increase in the irradiated thymus and take the place of CD4SP cells? To make this issue clear, the flow cytometry profiles and cell number data of LTi cells and other lymphoid cells including CD4SP cells before and after irradiation must be shown.

Response: We thank the reviewer for this constructive comment on the cell identity that overexpresses RANKL after irradiation. We fully agree with the fact that LTi cells constitute a rare cell type in the thymus even after irradiation (~0.09% or $3.35 \times 10^3 \pm 0.72 \times 10^3$ cells in untreated WT mice vs ~0.21% or $1.58 \times 10^3 \pm 0.12 \times 10^3$ cells after irradiation; *i.e.* a decrease of ~2 fold). In accordance with the reviewer comment and the study of Ueno *et al.*, (JEM; 2004), we observed that CD4⁺ thymocytes are partially radioresistant and thus constitute the main lymphoid population after irradiation (~10% or $18.01 \times 10^6 \pm 2.07 \times 10^6$ in untreated WT mice vs ~70% or $2.6 \times 10^6 \pm 0.2 \times 10^6$ cells; *i.e.* a decrease of ~7 fold). Therefore, LTi cells do not take the place of CD4⁺ thymocytes in the irradiated thymus. As requested by the reviewer, dot plots and numbers of LTi cells as well as those of other lymphoid cell populations before and after irradiation are now shown in Figure 1 B-C and Table 1 in this revised version. Concerning the identity of the cell population that upregulates RANKL after irradiation, we have used different genetically deficient mice to address this point (Cf. our response below).

The authors show that the RANKL expression after irradiation is significantly lower in Rorc^{-/-} thymus than in WT thymus (Fig. 1D), to claim that LTi cells are critical for RANKL up-regulation after thymic damage. However, this interpretation might be incorrect, because Rorc^{-/-} mice also exhibit severely reduced CD4SP cells (Science 288, 2369-2373, 2000). Thus, in addition to Rorc^{-/-} mice, Tcra^{-/-} mice should be used; Tcra^{-/-} mice have normal number of LTi cells but completely lack CD4SP cells, so if LTi cells were the main source of RANKL, they would show the high RANKL expression similar to WT.

Response: We fully agree that Rorc^{-/-} mice show reduced numbers of CD4⁺ thymocytes, as previously reported by Sun *et al.*, Science, 2000. To determine the contribution of CD4⁺ thymocytes in RANKL upregulation after irradiation, since Tcra^{-/-} mice are not directly available in our institute, we have used ZAP-70^{-/-} mice showing the same blockage in T-cell development at the DP stage and thus lacking SP thymocytes (Negishi I *et al.*, Nature. 1995 and Kadlecik *et al.*, J.Immunol, 1998). We found that ZAP-70^{-/-} thymi failed to increase RANKL expression after irradiation indicating that CD4⁺ thymocytes are critical for RANKL upregulation after SL-TBI (cf. Figure 1E). We also provide flow cytometry profiles showing that CD4⁺ thymocytes upregulated RANKL protein during the course of SL-TBI with no hematopoietic rescue but at a lesser extent than LTi cells (Figure 1D,F). Nevertheless, since LTi cells expressed high levels of RANKL after irradiation, we decided

to further determine the contribution of this cell type in RANKL upregulation. For this, we also analyzed the thymus of Rag2^{-/-} mice (lacking both DP and SP thymocytes but showing LT_i cells) and found that after irradiation, RANKL is upregulated in these mice but at lesser extent than in WT mice (cf. Figure 1E). Altogether, thanks to the reviewer comment, these new data indicate that CD4⁺ thymocytes are the main producers of RANKL after irradiation although that LT_i cells contribute in this process. The model recapitulating our new findings was thus accordingly modified (Cf. Appendix Figure S8).

The authors also concluded that LT_i cells are the main producer of LT α in irradiated thymus (Fig. 3), but this interpretation is also questionable, because LT α is also highly expressed in CD4SP cells.

Response: To investigate the role of CD4⁺ SP cells in LT α production after SL-TBI, we analyzed LT α expression in the thymus of irradiated ZAP-70^{-/-} mice, which upregulated LT α at the same level than that of the thymus of irradiated WT mice (cf. Figure 3F). These results thus show that in contrast to RANKL, CD4⁺ thymocytes are not involved in LT α overexpression after SL-TBI. However, since ZAP-70^{-/-} mice have DP and LT_i cells, we investigated the role of these two cell types in LT α overexpression after irradiation by using irradiated Rag2^{-/-} mice and irradiated WT mice treated three days before with dexamethasone, which specifically deplete DP cells (as described in Purton, *et al.*, J Immunol. 2004). We found that these two models expressed LT α at the same level than that of irradiated WT mice, showing that DP thymocytes are not implicated in LT α expression after SL-TBI (cf. Figure 3G). These different genetic approaches have thus confirmed that, though LT_i cells represent a rare cell type in the thymus, they constitute the main source of LT α after irradiation.

Second, the authors show that thymic LT_i cells express high level of RANK after irradiation, and the RANKLRANK signal induces in LT_i cells, LT α , which is critical for thymic regeneration. In view of this, it is worth to ask whether the LT α expression and thymic regeneration are impaired in the absence of RANKL, using RANKL-flox or RANK-flox mice crossed with lymphoid-specific Cre mice. Alternatively, the loss-of-function study on RANKL can be achieved if the authors use RANK-Fc protein. Whether the administration of RANK-Fc into irradiated mice could inhibit the LT α up-regulation and thymic regeneration must be determined.

Response: We thank the reviewer for this constructive comment. To analyze if LT α up-regulation and thymic regeneration are impaired in the absence of RANKL, a neutralizing anti-RANKL antibody (clone IK22/5; Miyahira Y, *et al.*, J. Immunol, 2003; Kamiyo S, *et al.*, Biochem. Biophys. Res. Commun, 2006) or a Rat IgG2a, κ isotype control was administrated *in vivo* during three consecutive days after SL-TBI. We found that the *in vivo* neutralization of RANKL fully inhibited LT α up-regulation selectively in LT_i cells. This result is now shown in Figure 3D,E. Furthermore, whereas RANKL *ex vivo* administration boosts TEC regeneration, conversely the neutralization of RANKL substantially impaired TEC recovery. These new results are shown in Figure 2A-D. We observed a decrease of around 2.5-fold in numbers of total TECs (EpCAM⁺), cTECs (EpCAM⁺UEA-1Ly51⁺) and mTECs (EpCAM⁺UEA-1⁺Ly51⁻) compared to control groups (cf. Figure 2A). mTEC subset analysis also revealed that CD80^{hi}Aire⁻ and CD80^{hi}Aire⁺ mTECs were substantially reduced after RANKL neutralization. TEC subsets identified on MHCII expression level, as previously reported by Wong *et al.* (Cell reports, 2014), namely cTEC^{hi} (MHCII^{hi}UEA-1⁻), mTEC^{hi} (MHCII^{hi}UEA-1⁺) and mTEC^{lo} (MHCII^{lo}UEA-1⁺) as well as TEPC-enriched cells were also diminished (cf. Figure 2B,C). Furthermore, RANKL neutralization also decreased numbers of proliferating Ki-67⁺ cTECs, mTECs and TEPC-enriched cells (cf. Figure 2D).

Altogether, thanks to the reviewer advice these new data indicate that RANKL plays a crucial role in endogenous TEC regeneration.

Lastly, it was also reported that RANKL stimulates osteoclasts in the bone to promote the mobilization of lymphoid progenitor cells (Nat. Med. 12, 657-664, 2006). The authors should examine and discuss this possibility.

Response:

We totally agree with the reviewer that the *in vivo* administration of RANKL protein in WT mice at steady state has been previously reported to promote the mobilization of lymphoid progenitors through the development of active osteoclasts. As requested by the reviewer, we examined in our

BMT experimental setting, the development of active osteoclasts by analyzing the expression of the phosphatase TRAP by histology (as previously described in Kollet *et al.*, Nat. Med. 2006) in the femurs of mice treated with GST or RANKL protein at day 2, 4 and 6 after BMT. TRAP staining was then performed at day 21 after BMT. We found an increase area of ~2-fold of TRAP⁺ osteoclasts in mice treated with RANKL compared to those treated with GST (cf. Appendix Figure S9A). Consistently with this observation, we further observed a ~2fold increase of primitive Lin⁻Sca-1⁺c-kit⁺ progenitors in the bone marrow of RANKL-treated animals (cf. Appendix Figure S9B). We therefore discuss the fact that RANKL could favor thymus homing of lymphoid progenitors and *de novo* thymopoiesis by increasing hematopoietic progenitor cells (cf. lines 413-416).

MINOR POINTS:

1. *Fig. 2H shows that the RANKL administration influences gene expression in cTECs. How can RANKL affect cTECs?*

Response: The reviewer is absolutely right, in Fig. 2H, we show that RANKL administration after SL-TBI increases in cTECs the expression of genes implicated in thymus homing. This could be explained by the fact that the administration of RANKL protein stimulates LT α upregulation in LTi cells as illustrated in Figure 3B, likely as a membrane-bound LT α 1 β 2 heterotrimer, and that cTECs overexpress the cognate LT β R receptor after irradiation as shown in Figure 4A. In accordance with these observations, it has been recently reported that LT β R regulates the expression of adhesion molecules implicated in T-cell progenitor homing (cf. Lucas B *et al.*, J Immunol. 2016). These data argue in favor of model in which RANKL acts indirectly on TECs through LT α overexpression in LTi cells. This point has been developed in the discussion of the revised version (cf. lines 464-467) and for a better clarity, a model recapitulating our findings is provided in Appendix Figure S8.

2. *In Fig. 5D, the authors examined the gene expression in CD45-thymic stromal cells and normalized the expression levels with EpCAM or CD31. However, CD45-thymic stroma contain not only TECs and endothelial cells but also other stromal cells such as fibroblasts, and some genes such as CCL25, CCL19 and CCL21 are also expressed in fibroblasts (J. Immunol., 178, 4956-4965, 2007). Thus, the data in Fig. 5D do not indicate the defective gene expression in TECs and endothelial cells.*

Response: We agree with the reviewer that CD45⁻thymic stroma contains not only TECs and endothelial cells but also fibroblasts. To remove any concerns, TECs, endothelial cells and fibroblasts were cell-sorted based on EpCAM, CD31 and gp38 markers, respectively (Cf. Figure 5D). We found that highly purified TECs from BM-transplanted LT α ^{-/-} mice showed a reduced expression of *Ccl19* and *Ccl21* mRNAs whereas purified fibroblasts displayed a reduced expression of *Ccl21* mRNA. Furthermore, purified endothelial cells exhibited a decrease expression of *Icam-1*, *Vcam-1* and *Selp* adhesion molecules. TECs, fibroblasts and endothelial cells are therefore defective in key molecules involved in thymus homing in BM-transplanted LT α ^{-/-} mice, which likely contributes in a defective thymus seeding of lymphoid progenitors (Cf. Figure 5E).

3. *Why did the authors need to perform BFA treatment and intracellular staining for detecting RANKL or LT α protein expression? Are those treatments needed for LT β R-Fc staining?*

Response:

We performed RANKL intracellular staining to detect the presence of both intracellular and membrane-bound forms. We are sorry for the confusion in the methods since BFA was not used for RANKL staining. In the revised version, we have corrected this mistake in the Material and Methods.

For LT α staining, since LT α exists as a secreted LT α 3 homotrimer and as a membrane anchored LT α 1 β 2 heterocomplex (Gommerman JL and Browning JL, Nat Rev Immunol, 2003), thymic cells were first treated with BFA that inhibits protein secretion in order to detect both membrane-bound and soluble forms. Intracellular staining was performed as recommended by the manufacturer's instructions.

For LT β R-Fc, staining was performed without BFA treatment and cell surface staining was performed on ice to avoid any potential internalization of LT α 1 β 2 heterotrimer, as previously described in Boehm T *et al.*, JEM. 2003.

4. In Fig. 2A and Fig. 4B, flow cytometry profiles for total TECs in total live cells are shown. It is supposed that these are not total live thymic cells but likely MACS-enriched or CD45-negative stromal cells. Must be described in the figure legends.

Response: The reviewer is absolutely right. In Fig. 2A and Fig. 4B, TECs were analyzed in CD45^{neg}-enriched cells by AutoMACS and not in total live cells. We thus replaced “total live cells” by “CD45^{neg}-enriched cells” in these two figures. The figure legends were accordingly modified.

2nd Editorial Decision

07 March 2017

Thank you for the submission of your revised manuscript to EMBO Molecular Medicine. We have now received the enclosed reports from the referees that were asked to re-assess it. As you will see the reviewers are now globally supportive and I am pleased to inform you that we will be able to accept your manuscript pending the following final amendments:

1) Please address the minor change commented by referee 2. Please provide a letter INCLUDING the reviewer's reports and your detailed responses to their comments (as Word file).

Please submit your revised manuscript within two weeks. I look forward to seeing a revised form of your manuscript as soon as possible.

***** Reviewer's comments *****

Referee #1 (Remarks):

The authors answered properly to all my comments.

Referee #2 (Remarks):

The additional experiments described in the revised version of the manuscript are well performed. The loss-of-function studies using a neutralizing anti-RANKL antibody clearly demonstrate the pivotal role of RANKL signaling in LTa up-regulation and thymic regeneration. The authors also clarified that CD4SP thymocytes are the main RANKL producers after irradiation but LTi cells play a dominant role in the regeneration process. Thus our criticisms have been properly addressed in the revised version of manuscript, although only the following minor concern remains.

As far as I know, no reports have shown that bisphosphonates inhibit osteoclast 'differentiation'. Tomimori et al., 2009 demonstrated an inhibitory effect of bisphosphonates on bone resorption induced by RANKL administration but they have not shown the effects on osteoclast differentiation. To avoid confusion, the description about osteoclast 'differentiation' (lines 484 and 487) should be removed.

2nd Revision - authors' response

22 March 2017

We have addressed the minor issue raised by Reviewer 2 by removing the description about osteoclast differentiation.

Corresponding Author Name: Magali IRLA

Journal Submitted to: Embo Molecular Medicine

Manuscript Number: EMM-2016-07176